



# A low-cost monitor for simultaneous measurement of fine particulate matter and aerosol optical depth – Part 3: Automation and design improvements

Eric A. Wendt[1], Casey Quinn[1], Christian L'Orange[1], Daniel D. Miller-Lionberg[3], Bonne Ford[4], Jeffrey R. Pierce[4], John Mehaffy[1], Michael Cheeseman[4], Shantanu H. Jathar[1], David H. Hagan[6], Zoey Rosen[5], Marilee Long[5], and John Volckens[1,2]

[1]Department of Mechanical Engineering, Colorado State University, Fort Collins, Colorado, USA, 80523

[2]Department of Environmental and Radiological Health Sciences, Colorado State University, Fort Collins, USA, 80523

[3]Access Sensor Technologies, LLC, Fort Collins, Colorado, USA, 80523

[4]Department of Atmospheric Science, Colorado State University, Fort Collins, Colorado, USA, 80523

[5]Department of Journalism and Media Communication, Colorado State University, Fort Collins, Colorado, USA,

15  80523

[6]QuantAQ, Inc., Somerville, Massachusetts, USA, 02143

*Correspondence to*: John Volckens (john.volckens@colostate.edu)

**Abstract.**

Atmospheric particulate matter smaller than 2.5 microns in diameter ($PM_{2.5}$) impacts public health, the environment, and the climate. Consequently, a need exists for accurate, distributed measurements of surface-level $PM_{2.5}$ concentrations at a global scale. Remote sensing observations of aerosol optical depth (AOD) have been used to estimate surface-level $PM_{2.5}$ for studies on human health and the Earth system. However, these estimates are uncertain due to a lack of measurements available to validate the derived $PM_{2.5}$ products, which rely on the ratio of

surface $PM_{2.5}$ to AOD. Traditional monitoring of these two air quality metrics is costly and cumbersome, leading to a lack of surface monitoring networks with high spatial density. In part 1 of this series we described the development and validation of a first-generation device for low-cost measurement of AOD and $PM_{2.5}$: The Aerosol Mass and Optical Depth (AMODv1) sampler. Part 2 of the series describes a citizen-science field deployment of the AMODv1 device. Here in part 3, we present an autonomous version of the AMOD, known as AMODv2, capable of

unsupervised measurement of AOD and $PM_{2.5}$ at 20-minute time intervals. The AMODv2 includes a set of four optically filtered photodiodes for multi-wavelength (current version at 440, 500, 675, and 870 nm) AOD, a Plantower PMS5003 sensor for time-resolved optical $PM_{2.5}$ measurements, and a pump and cyclone system for time-



integrated gravimetric filter measurements of particle mass and composition. The AMODv2 uses low-cost motors
and sensor data for autonomous sun alignment to provide the semi-continuous AOD measurements. Operators can
connect to the AMODv2 over Bluetooth® and configure a sample using a smartphone application. A Wi-Fi module
enables real-time data streaming and visualization on our website (csu-ceams.com). We present a sample
deployment of 10 AMODv2s during a wildfire smoke event and demonstrate the ability of the instrument to capture
changes in air quality at sub-hourly time resolution. We also present the results of an AOD validation campaign
where AMODv2s were co-located with AERONET (Aerosol Robotics Network) instruments as the reference
method at AOD levels ranging from 0.016 to 1.59. We observed close agreement between AMODv2s and the
reference instrument with mean absolute errors of 0.046, 0.057, 0.026, and 0.033 AOD units at 440 nm, 500 nm, 675
nm, and 870 nm, respectively. We identified individual unit bias as the primary source of error between AMODv2s
and reference units and propose re-calibration to mitigate these biases. The AMODv2 is well suited for citizen-
science and other high-spatial-density deployments due to its low cost, compact form, user-friendly interface, and
high measurement frequency of AOD and $PM_{2.5}$. These deployments could provide a rich air pollution data set for
evaluating remote sensing observations, atmospheric modeling simulations, and provide communities with the
information they need to implement effective public health and environmental interventions.

## 1 Introduction

Fine particulate matter air pollution ($PM_{2.5}$) is a leading cause of morbidity, mortality, and environmental
change worldwide (Myhre et al., 2013; Forouzanfar et al., 2016; Brauer et al., 2016; Vohra et al., 2021). Inhaled
$PM_{2.5}$ can penetrate deep into the lungs, leading to both acute and chronic health impacts (Pope and Dockery, 2006;
Janssen et al., 2013; Feng et al., 2016; Kim et al., 2019). Each year, millions of deaths worldwide are attributed to
$PM_{2.5}$ exposure (Brauer et al., 2016; Forouzanfar et al., 2016). In addition to public health, $PM_{2.5}$ also contributes to
visual degradation of the atmosphere and affects the climate by influencing Earth's radiative budget (Myhre et al.,
2013). Regions with the highest levels of air pollution often lack adequate ground level monitoring (Snider et al.,
2015; Brauer et al., 2016). Thus, disease estimates for much of the world's population rely on exposure estimates
where satellite data or model simulations are the best or only source of information on human exposure. Installing a
global network of reference-grade surface monitors is not currently feasible due to the high installation and
maintenance costs.

Satellite remote sensing, supplemented with data from surface monitors and chemical transport models
(CTMs), represents the state-of-the-art for global $PM_{2.5}$ monitoring at relatively high temporal and spatial resolution
(van Donkelaar et al., 2016, 2019; Hammer et al., 2020; Lee, 2020). Measurements from satellite instruments, such
as the Moderate Resolution Imaging Spectrometer (MODIS) and the Multi-angle Imaging SpectroRadiometer
(MISR) (Salomonson et al., 1989; Diner et al., 1998), are used to estimate surface-level $PM_{2.5}$ concentrations (e.g
Liu et al., 2005), which in turn have facilitated research on the health effects associated with $PM_{2.5}$ exposure (Brauer
et al., 2016; Forouzanfar et al., 2016; Li et al., 2018; Lu et al., 2019). Satellites equipped with specialized
instruments retrieve aerosol optical depth (AOD), a parameter related to light extinction in the atmospheric column,
which can then be converted to ground level $PM_{2.5}$ using a CTM or statistical relationship (Liu et al., 2005; van



Donkelaar et al., 2006, 2010, 2012, 2016; Hammer et al., 2020). The relationship between AOD and $PM_{2.5}$ can be
expressed as follows (Liu et al., 2005):

$$PM_{2.5} = \eta \cdot AOD \tag{1}$$

where η is a conversion factor between $PM_{2.5}$ and AOD. The uncertainty of surface-level $PM_{2.5}$ concentrations
derived from satellite observations has two main components: 1) the uncertainty of the satellite AOD measurement
and 2) the uncertainty of the modeled $PM_{2.5}$ to AOD ratio (η) (e.g. Ford and Heald, 2016; Jin et al., 2019).

The error of the satellite AOD retrieval can be estimated using ground-level AOD measurements from
instruments known as sun photometers (e.g., Sayer et al., 2012). The Aerosol Robotics Network (AERONET)
provides reference-quality AOD measurements at hundreds of locations around the Earth; these data are used to
constrain and reduce uncertainties in AOD values (Holben et al., 1998). AERONET instruments are rarely deployed
at high spatial density (i.e. sub-city scale), outside of field campaigns (e.g. Garay et al., 2017), due to the high cost
of the instrument and supporting equipment (>$50,000). Determining the uncertainty in the modeled $PM_{2.5}$ to AOD
ratio requires co-locating AOD monitors with $PM_{2.5}$ monitors. The Surface PARTiculate mAtter Network
(SPARTAN) was established to provide co-located $PM_{2.5}$ and AOD reference measurements and to evaluate
uncertainties in both AOD and the $PM_{2.5}$ to AOD ratio; however, the number of SPARTAN sites worldwide is
limited by number (~20 active sites), equipment and operational costs (Snider et al., 2015).

85        Networks of low-cost nephelometers (notably the Plantower PMS5003), have been suggested and deployed
in large numbers as a means to provide surface $PM_{2.5}$ validation data at a higher spatial density than can be achieved
with high-cost monitors (Lin et al., 2020; Li et al., 2020; Badura et al., 2020; Lu et al., 2021; Chadwick et al., 2021).
However, low-cost sensors (or more specifically, the Plantower PMS5003 devices) tend to exhibit measurement bias
(Kelly et al., 2017; Zheng et al., 2018; Levy Zamora et al., 2019; Sayahi et al., 2019; Tryner et al., 2020), requiring
correction relative to reference monitors (Ford et al., 2019; Wendt et al., 2019). Low-cost Sun photometers have
been deployed at high-spatial resolution to evaluate satellite AOD uncertainty as part of the Global Learning and
Observations to Benefit the Environment (GLOBE) program (Boersma and de Vroom, 2006; Brooks and Mims,
2001). GLOBE Sun photometers were operated by students as part of education programming, resulting in over 400
measurements between January 2002 and October 2005 in the Netherlands (Boersma and de Vroom, 2006). These
data were used to evaluate satellite-derived AOD in corresponding regions. However, the authors noted difficulty
coordinating with schools to achieve consistent measurements, specifically those corresponding with satellite
overpasses. Collectively, these previous efforts have advanced the understanding of AOD and $PM_{2.5}$:AOD
variability considerably. However, there is still demand for co-located $PM_{2.5}$ and AOD samplers deployed at higher
spatial density and with greater temporal resolution (Ford and Heald, 2016; Garay et al., 2017; Jin et al., 2019).
Samplers used in these networks must be sufficiently low-cost to deploy in large numbers, have manageable
operational and maintenance requirements, and provide accurate $PM_{2.5}$ and AOD measurements.

       In part 1 of this series of articles, we developed a low-cost, compact $PM_{2.5}$ and AOD ground monitor
(Wendt et al., 2019; Ford et al., 2019). The device, known as the Aerosol Mass and Optical Depth (AMOD)
sampler, featured a $PM_{2.5}$ cyclone inlet for integrated gravimetric sampling and composition analysis, a low-cost
nephelometer (Plantower PMS5003, Beijing, China) for real-time $PM_{2.5}$ mass estimate, and four filtered-photodiode





(Intor Inc., Socorro, NM, USA) sensors at 440, 520, 680, and 870 nm for measuring AOD. Here, we refer to this earlier instrument as the AMODv1. The assembly cost for the first manufacturing set of 25 AMODv1s was under $1,100 per unit, less than 1/60[th] the combined purchase prices of reference AOD and $PM_{2.5}$ monitors (Wendt et al., 2019). The results of a field validation campaign revealed agreement to within 10% (mean relative error) for AOD

values relative to co-located AERONET monitors and for $PM_{2.5}$ values relative to Federal Equivalent Method (FEM) monitors from the Environmental Protection Agency (EPA) (Wendt et al., 2019). These results indicated that the AMODv1 accurately quantified surface $PM_{2.5}$ concentrations and AOD simultaneously and at a substantially lower cost and smaller size than existing equipment. To test implementation of the AMODv1, we constructed and deployed 25 AMODv1s in a citizen-science network, as documented in part 2 in this series (Ford et al., 2019).

Despite the promise of the AMODv1, the initial deployment highlighted several key limitations. First, the AMODv1 lacked quality control measures for misalignment or cloud contamination during the measurement period. Second, the instrument had limited temporal resolution for AOD (typically one measurement per day). Third, despite the presence of a visual alignment aid (Wendt et al., 2019), many volunteers found it difficult to align the instrument with the sun, which was compounded by inconsistent standards as to what constituted proper alignment.

Fourth, data could not be transmitted wirelessly or accessed remotely. Hence, the primary objective of this current work was to design and integrate a system for automatic multi-wavelength AOD measurements throughout daylight hours and to validate the performance of this system against AERONET reference monitors. We integrated this technology with real-time and gravimetric $PM_{2.5}$ sampling technology to produce a low-cost and mobile instrument capable of acquiring extensive surface and columnar air quality data at a lower cost. The AMODv2 was designed to

automate the AOD measurement and data transfer functionality, while integrating systems to measure both time-integrated (gravimetric, filter-based) and real-time $PM_{2.5}$. Here, we describe the design and validation of the AMODv2.

## 2 Materials and methods

### 2.1 Instrument design

We designed the AMODv2 to sample integrated gravimetric $PM_{2.5}$ mass concentration, real-time $PM_{2.5}$ mass concentration, and AOD simultaneously. One intended application is large-scale sampling campaigns with the AMODv2 instruments operated by volunteers with little to no background in aerosol or atmospheric science (Ford et al., 2019). Thus, we prioritized a design that is low-cost, mechanically robust, portable, automated, and user-friendly. We provide images of AMODv2 hardware in Fig. 1, highlighting key internal and external components.



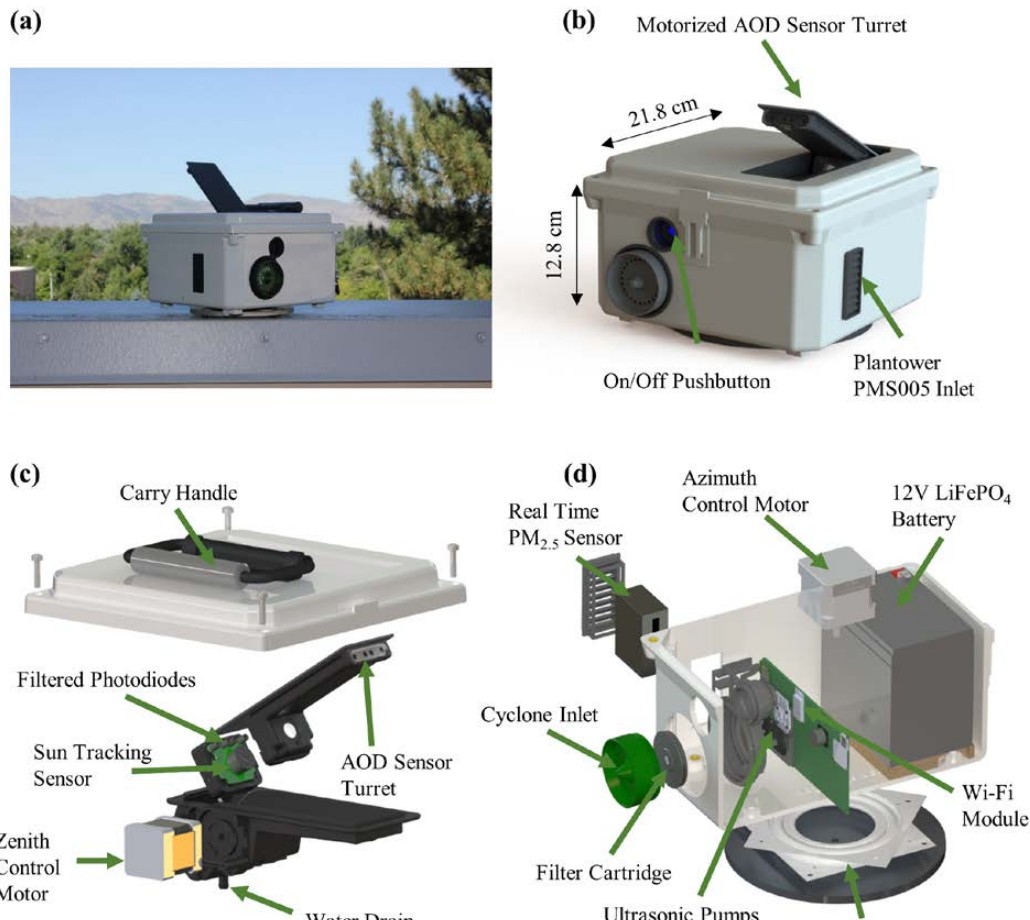

**Figure 1: Images detailing external and internal AMODv2 design and hardware. a) Photograph of AMODv2 sampling outdoors. b) External computer animated rendering of AMODv2 features and dimensions. c) Computer generated exploded view of AOD measurement subsystem. d) Computer generated exploded section view of PM$_{2.5}$ sampling, wireless data transfer, and power subsystems.**



The AMODv2 measures AOD at 440 nm, 500 nm, 675 nm, and 870 nm using optically filtered photodiodes (Intor Inc., Socorro, NM, USA) with narrow bandwidth (<15 nm at full-width half-maximum signal). The measurement process is fully automated using a solar tracking system (Section 2.3), reducing the potential for misalignment due to user error. Movement in the zenithal plane is achieved using a custom turret module embedded

in the interior of the AMODv2 enclosure (Fig. 1a). The module was designed in SolidWorks® (ANSYS, Inc., Canonsburg, PA, USA) and built using multi-jet fusion printing. The module houses a custom printed circuit board containing the solar tracking sensors and the filtered photodiodes. Light enters the turret through four, 4 mm



apertures, and passes through 112 mm tubes to reach the filtered photodiodes (Fig. 1c). These proportions yield a viewing angle of approximately 2 degrees for each photodiode sensor element. A stepper motor (Stepper Online 17HS10-0704S-C2, Nanjing City, China), fixed to the turret, actuated the zenithal rotation. Movement in the azimuthal plane is actuated using a second stepper motor (Stepper Online 17HS19-1684S-C6, Nanjing City, China) fixed to a turntable and base-plate assembly (McMaster Carr 6031K16, Elmhurst, IL, USA), which enables 360 degree rotation of the AMODv2. The angular resolution of each stepper motor is tuned to 0.056 degrees using programmable drivers (Texas Instruments DRV8834RGER, Dallas, Texas, USA). Active tracking is accomplished using closed-loop control enabled by a 3-axis accelerometer (STMicroelectronics LSM6DSM, Geneva, Switzerland), a GPS module (u-blox CAM-M8, Thalwil, Switzerland), and a quadrant photodiode solar tracking sensor (Solar MEMS NANO-ISS5, Seville, Spain).

The AMODv2 measures $PM_{2.5}$ using both real-time and time-integrated techniques. Real-time $PM_{2.5}$ concentrations are measured and streamed using a light-scattering $PM_{2.5}$ sensor (Plantower PMS5003, Beijing, China). A 3D-printed fixture secured the sensor in position to sample ambient air, while downward sloping vents protect the sensor from water ingress (Fig. 1d). $PM_{2.5}$ concentrations are evaluated on the PMS5003 chip via a manufacturer proprietary algorithm. The AMODv2 reports the $PM_{2.5}$ values corrected by Plantower's proprietary atmospheric correction. These values are accessed by the AMODv2 microcontroller via serial communication. A flow chart detailing the $PM_{2.5}$ measurement protocol is provided in Fig. S1.

For time-integrated $PM_{2.5}$ mass concentration measurement, we leveraged a $PM_{2.5}$ cyclone design from prior studies (Volckens et al., 2017; Kelleher et al., 2018; Wendt et al., 2019). The main circuit board features ultrasonic pumps (Murata MZBD001, Nagaokakyo, Japan) and a mass flow sensor (Honeywell Omron D6F, Charlotte, NC, USA,) to control the flow of air through a custom aluminum cyclone and filter cartridge with a 50% cut point of 2.5 μm (Fig. 1d). The gravimetric sample is collected on a 37mm Teflon filter secured within a filter cartridge. Sampled particles are collected on a single filter that is pre and post weighed for each sample. During deployment, a field blank is carried along with the sampler to correct for incidental mass contamination.

The AMODv2 is powered using a 12 V, 10 Ah $LiFePO_4$ battery (Dakota Lithium, Grand Fork, ND, USA) with a secondary 12 V, 3.3Ah $LiFePO_4$ (Battery Space, LFH4S4R1WR-C5, Richmond, CA, USA) battery in parallel. The battery is charged using a barrel plug inlet accessible on the side of the enclosure. A detachable rubber plug seals the inlet from the outside environment when not charging. Charging circuitry supports charging at a rate of 3.0 A, enabling a full charge in approximately eight hours. A full charge can power the AMODv2 for over 120 hours.

The AMODv2 records and wirelessly transfers meteorological and quality-control data in real time. Meteorological data include ambient temperature (°C), ambient pressure (hPa), and relative humidity (%). Quality control metrics include sample duration (s), sample flow rate (L min$^{-1}$), total sampled volume (L), battery temperature (°C), battery voltage (V), battery state of charge (%), current draw (mA), and wireless signal strength (RSSI).

The external housing of the AMODv2 (Fig. 1b) is made from a weather-resistant NEMA electrical enclosure (Polycase, YQ-080804, Avon, Ohio, USA). The dimensions of a fully assembled AMODv2 are 21.8 cm





W × 21.8 cm L × 12.8 cm H, with a weight of 3.1 kg. A folding carry handle is fixed to the upper surface of the
enclosure to aid transport (Fig. 1b). The total cost of the AMODv2 was $1,175 per unit, for a production run of 100
units (Table S1). This tabulation includes an estimated three hours of assembly at a rate of $25 per hour.

    We developed the AMODv2 control software using an online, open-source platform (mbed™; ARM® Ltd.,
Cambridge, UK). The software was written in C++ and executed by a 64-bit microcontroller (STMicroelectronics

STM32L476RG, Geneva, Switzerland). We implemented wireless data transfer using a Wi-Fi and Bluetooth™
module (Espressif Systems ESP32-C3-WROOM, Shanghai, China). A MicroSD card stores all data for data backup
or offline deployment (Molex 5031821852, Lisle, IL, USA). We integrated software modules for AOD, real-time
$PM_{2.5}$, gravimetric $PM_{2.5}$, data logging, and wireless data transfer using a real-time operating system (RTOS) for
pseudo-simultaneous software execution.

**2.2 AOD measurement and solar tracking**

    The AMODv2 applies the Beer-Lambert-Bouguer law to calculate AOD ($\tau_a$). This relationship, expressed
in terms of measurable parameters, is as follows:

$$\tau_a(\lambda) = \frac{1}{m}\left(ln\left(\frac{V_0}{R^2}\right) - ln(V)\ \right) - \tau_R(\lambda, p) - \tau_{O3} \tag{2}$$

where *m* is the unitless air mass factor, which accounts for the increased air mass that light passes through as the sun

approaches the horizon, R is the Earth-sun distance in astronomical units (AU), V is the signal produced by the light
detector in volt, $\tau_R$ accounts for Rayleigh scattering by air molecules, p is the pressure at the sensor in Pa, λ is the
sensor wavelength in m, $\tau_{O3}$ accounts for ozone absorption, and $V_0$ is the extraterrestrial constant in volt, which is
the sensor signal if measured at top-of-atmosphere and is determined via calibration. AOD values at 440 nm, 500
nm, 675 nm, and 870 nm are calculated using Eq. (2). The Earth-Sun distance, R, is computed directly from GPS

data and the solar positioning algorithm. V is the signal produced by the photodiode and $V_0$ is accessed from on-chip
memory. The relative optical air mass factor is computed as a function of solar zenith angle (θ) as follows (Young,
1994):

$$m\ =\ \frac{1.002432 \cdot cos^2(\theta) + 0.148386 \cdot cos(\theta) + 0.0096467}{cos^2(\theta) + 0.149864 \cdot cos^2(\theta) + 0.0102963 \cdot cos(\theta) + 0.000303978} \tag{3}$$

    The contribution of total optical depth from Rayleigh scattering, $\tau_R$, is calculated based on wavelength and

ambient pressure measured by an ambient pressure sensor mounted on the circuit board with the AOD sensors
(Bosch Sensortec BMP 280, Kusterdingen, Germany) (Bodhaine et al., 1999). Ozone concentration is estimated
using an empirical model based on time of year and location, and converted to $\tau_{O3}$ using wavelength-specific ozone
absorption coefficients (Griggs, 1968; Van Heuklon, 1979). With all parameters known, Eq. (2) is applied to
calculate AOD.

We implemented automatic solar tracking capabilities using a suite of low cost sensors and a multi-stage
algorithm. Detailed flow charts of the AOD measurement protocol are provided in the supplementary Figs. S2-S6.
The open loop stage is initiated when the microcontroller requests an AOD measurement and the GPS time and
location is computed. Using this information, the AMODv2 applies a solar positioning algorithm from the National
Renewable Energy Laboratory (NREL) to compute the solar elevation angle (Reda and Andreas, 2008). The





calculated solar zenith angle is then compared with the pitch of the AOD turret relative to horizontal. The turret stepper motor rotates the turret in the direction of the sun until the elevation angle of the AOD turret is approximately equal to that of the sun. The base motor rotates counterclockwise in order to achieve approximate azimuth alignment. After every 10 degrees of azimuthal rotation, the total signal of the sun-tracking quadrant photodiode is compared with an empirical threshold. If the threshold is exceeded, the AMODv2 enters closed-loop

tracking. If the threshold is not exceeded on the first revolution, the AMODv2 executes a second revolution before ending the search protocol.

In the closed-loop tracking stage, the rotation of the motors is controlled using the zenithal and azimuthal error signals produced by the quadrant photodiode. The quadrant photodiode is mounted in a diamond orientation, with two quadrants forming a vertical axis, and two forming a horizontal axis. The vertical error signal is the ratio of

the top and bottom quadrants and the horizontal error signal is the ratio of the right and left quadrants. The stepper motors rotate independently until each error signal is reduced within an experimentally determined threshold. The motors then lock in place while an AOD measurement is recorded. The AMODv2 measures AOD as triplet sets. Between each measurement, both motors disengage for 30 seconds to conserve power. After 30 seconds, the AMODv2 executes the tracking algorithm and records an AOD measurement. This process is repeated until the

triplet set is completed or until 3 minutes have elapsed since the initial measurement request was made by the processor.

Real-time quality control is performed on each measurement triplet. Empty or incomplete triplets are flagged and assigned an error code. Completed triplets are screened for cloud contamination using the AERONET triplet cloud screening algorithm (Smirnov et al., 2000; Giles et al., 2019). The algorithm takes the maximum

deviation of any two measurements within a triplet, and applies thresholds to mark triplets as clear or cloud-contaminated (Smirnov et al., 2000; Giles et al., 2019). Large deviations of AOD within a triplet are more likely due to cloud contamination than changes in aerosol loading (Smirnov et al., 2000; Giles et al., 2019). Measurements identified as cloud-contaminated are flagged with a unique error code.

### 2.3 AOD calibration procedure

The extraterrestrial constants for all AMODv2s were evaluated via calibration relative to AERONET sun photometers (Cimel CE318, Paris, France) (Holben et al., 1998). AERONET monitors report AOD at 340 nm, 380 nm, 440 nm, 500 nm, 675 nm, 870 nm, 1020 nm, and 1640 nm (Holben et al., 1998). We selected the four AMODv2 AOD wavelengths in part for direct comparison with AERONET monitors. We conducted calibrations at the MAXAR-FUTON site in Fort Lupton, Colorado (40.036 N, 104.885 W) between November 2019 and February

2020. AMODv2 units were co-located within 50 m of the AERONET monitor and sampled for 2 to 3 hours at a rate of one sample every 2.5 to 3 minutes. AMODv2 and AERONET level 1.0 measurements concurrent within 60 seconds of each other were included in the calibration data set (Holben et al., 1998). For each set of concurrent measurements, we calculated the extraterrestrial constant by applying Eq. (2) solved for $V_0$, where $V$ was the raw voltage reported by the AMODv2, and $\tau_a$ was the AOD reported by the AERONET monitor. The AMODv2

calibration constants were the average value of $V_0$ for a given instrument and wavelength.





### 2.4 User operation and measurement procedure

We designed the AMODv2 to be operated by individuals without a background in aerosol sampling. We developed a standard procedure that is detailed in a user manual provided as supplementary material. After the initial setup, the AMODv2 requires no operator inputs for the duration of the sample. A flow chart outlining the manual and automatic steps to perform an AMODv2 measurement is provided in Fig. 2.


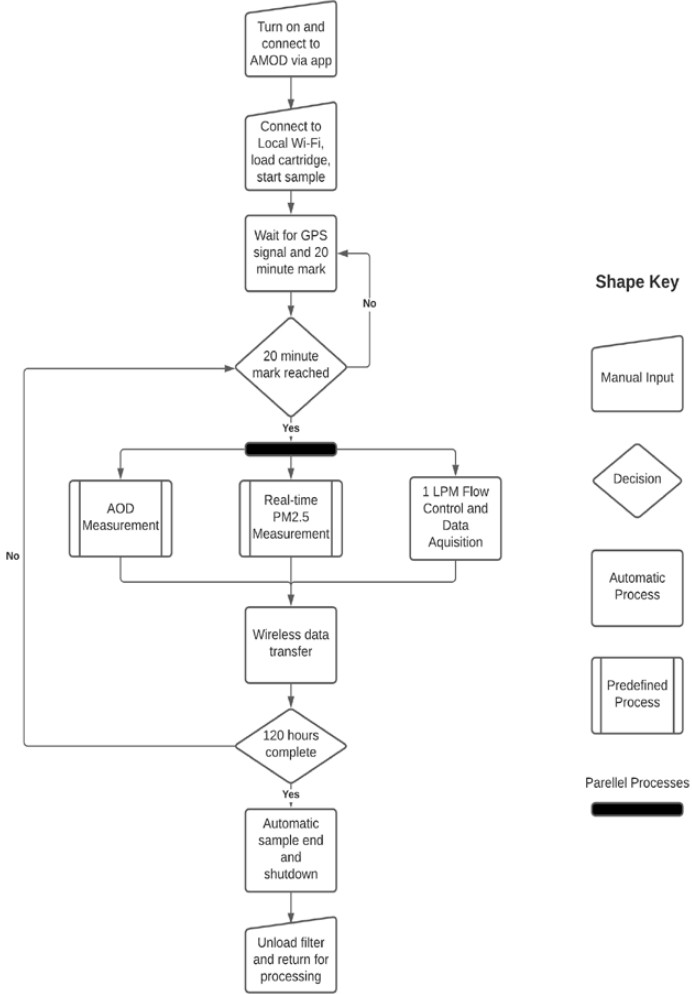

**Figure 2: Overall device operation flow diagram. Manual inputs require operator intervention. Automatic processes are executed with no operator intervention. Predefined processes are detailed in Supplemental Figs. S1-S6. Parallel processes are executed pseudo-simultaneously using a real-time operating system.**


To initiate a sample, the operator needs only an AMODv2 monitor, a cartridge loaded with a pre-weighed filter, and a smartphone with the AMODv2 control application installed ("CEAMS"; available on the Apple App



Store and Google Play) (Quinn 2019). A detailed description of the mobile application is in the user manual, which is included as a supplement to this work. After executing an initialization routine by selecting "Scan for Device", the

operator may connect to their device via Bluetooth™ using the mobile application. The operator can select a wireless network and input the proper credentials to connect the AMODv2 to the internet. The application then prompts the operator to scan the QR code on the back of the filter cartridge to link the filter with the upcoming sample in the data log. After the cartridge is manually loaded into the compartment behind the inlet (Fig. 1b), the AMODv2 should be placed on a flat surface with an unobstructed view of the sun. The operator then starts the

sample from the mobile application. After an initial data push, the sample begins at the next 20 minute mark (e.g. 12:00, 12:20, or 12:40). The AMODv2 begins sampling air through the inlet at 1 L min$^{-1}$ and continues to do so for the remainder of the 120-hour sampling period. Real-time $PM_{2.5}$ and AOD measurements are initiated at each 20 minute mark from the start of the sample. The $PM_{2.5}$ reported at each 20 minute interval is the average of measurements taken every 10 seconds over a period of 3 minutes. If the sun is less than 10 degrees above the

horizon, the motors do not activate and the solar tracking algorithm is not executed. After each AOD and $PM_{2.5}$ measurement is completed, data are uploaded to the affiliated website (csu-ceams.com), where real-time visualizations of AOD and $PM_{2.5}$ are available. Data reported to the website are accessible with a map-based user interface. Quality-control data are available to research staff via a private administrator portal. A snapshot example of the website is provided in Fig. S7. At the conclusion of a sample, the operator removes the filter cartridge. Upon

receipt of the filters, the CEAMS team stored the filters in the refrigerator until mailed to minimize loss of volatile compounds. Complete data files can be downloaded from the website or accessed via a MicroSD card.

### 2.5 Sample deployment and AOD validation studies

We conducted a sample deployment of 10 AMOD units during a wildfire smoke event in Fort Collins, Colorado in October of 2021. We configured the units to sample for approximately 60 hours. The 10 units were co-

located and sampled simultaneously. We collected and analysed real-time $PM_{2.5}$ mass concentrations, AOD, $PM_{2.5}$ to AOD ratio, meteorological data, and quality control data.

We assessed precision and bias of AMODv2 AOD sensors relative to an AERONET monitor at the NEON-CVALLA site in Longmont, Colorado (40.160 N, 105.167 W) between June 2020 and December 2020. We co-located our instruments within 50 m on seven separate days with varying atmospheric conditions (e.g. wildfire

smoke and clean air) using a total of 14 unique AMODv2s. Each test consisted of 2 to 4 hours of sampling at a rate of one sample every 2.5 to 3 minutes. AMODv2 and AERONET measurements concurrent within 120 seconds were included in the validation data set. The accuracy of AMODv2 AOD measurements was assessed via Deming regression analysis.

Compared with our prior work (Wendt et al., 2019), we tested the AMODv2 AOD measurement system

under a broader range of atmospheric conditions. A sizable portion of validation measurements were taken under heavy smoke caused by the Cameron Peak and East Troublesome fires of 2020. We conducted additional testing under more moderate smoke and clear conditions. AOD values reported by AERONET during validation



experiments ranged from 0.035 to 1.59 at 440 nm, 0.030 to 1.51 at 500 nm, 0.021 to 1.13 at 675 nm, and 0.016 to 0.77 at 870 nm.

**3 Results and discussion**

**3.1 Sample AMODv2 deployment**

The AMODv2 is capable of accurate, real-time, and low-cost measurement of AOD and $PM_{2.5}$. Here we present results from the sample deployment of 10 units. In Fig. 3, we provide real-time AOD at 500 nm, real-time $PM_{2.5}$, and the corresponding $PM_{2.5}$ to AOD ratios.

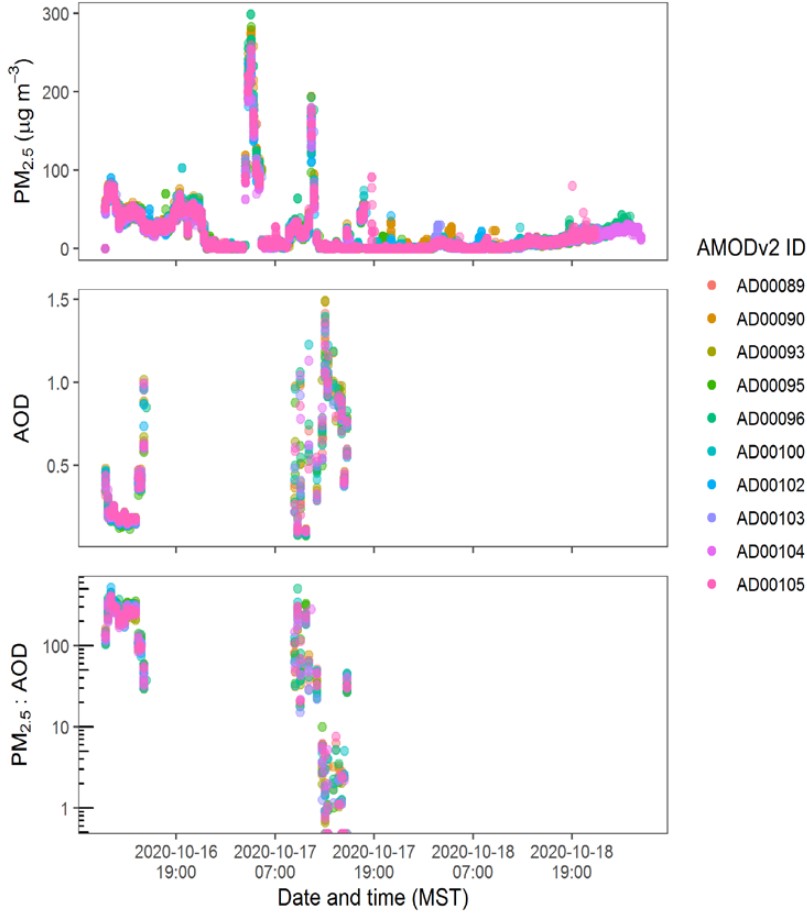


**Figure 3: Time series from 10 co-located AMODv2s featuring $PM_{2.5}$ concentration, AOD at 500 nm, and $PM_{2.5}$ to AOD (at 500nm) ratio for 17-19 October 2020 in MST. Points are colored according to the AMODv2 ID. Note the vertical axis for $PM_{2.5}$:AOD is provided in a logarithmic scale to clarify lower values indicative of lofted smoke.**


In supplemental Fig. S8, we provide detailed results from a single unit including 4-channel AOD, $PM_{2.5}$, and meteorological data including temperature, pressure, and relative humidity. This sample deployment highlighted several important strengths of the AMODv2 relative to AMODv1 and other prior sampling approaches. The high temporal resolution of AOD and $PM_{2.5}$ measurements facilitated a more complete understanding of the air pollution event occurring during the sample. With the AMODv2, we observed moderate air pollution at the start of the sample

on the afternoon of 16 October 2020, with all units reporting consistent values for AOD (>0.3) and $PM_{2.5}$ (50 to 100 $\mu g\ m^{-3}$). This was followed by increases on 17 October 2020 to severe levels (AOD up to 1.5 and PM2.5 up to 300 $\mu g\ m^{-3}$) as wildfire smoke swept over the city in the afternoon and gradually subsided over the course of 18 October 2020. We observed reductions in $PM_{2.5}$:AOD (<10) as ground level $PM_{2.5}$ decreased to moderate and mild levels (<20 $\mu g\ m^{-3}$), while the AOD remained elevated (>0.5) due to the presence of lofted smoke. We then noted the

continuation of the trend at ground level with the further reduction of ground-level $PM_{2.5}$ on 19 October 2020 (5 to 15 $\mu g\ m^{-3}$). Cloud cover prevented additional AOD measurements on 19th October, which was automatically screened for using the cloud screening algorithm. The meteorological data was also consistent with cloud cover with lower temperatures and elevated relative humidity reported on that day (Fig. S8).

       Data from the sample deployment were accessed from our companion website (csu-ceams.com) in real

time. With AOD, $PM_{2.5}$ and $PM_{2.5}$:AOD reported every 20 minutes throughout the sample to the website, we could assess the progress of wildfire smoke in Fort Collins remotely in real time. This was not possible with AMODv1, which lacked wireless transmission capabilities. In terms of scalability, the AMODv2 was relatively easy to deploy and maintain owing to its compact and weatherproof design, coupled with its automated measurement protocols. In the sample test, we were able to quickly prepare and deploy units in response to wildfire activity.

We leveraged the data accessibility features of AMODv2 for real-time quality control of incoming sample data. We monitored sample flow rate and total sampled volume to detect potential errors with the gravimetric sample collection. We monitored battery temperature to detect potential overheating of the unit, allowing proper intervention (e.g. temporarily moving the unit into shade) before the instrument reaches a shutoff threshold. We used battery voltage, battery state of charge, and current draw data to identify units unlikely to complete the intended

sample duration. Current draw data was also used to identify when the tracking motors were engaged, indicating an attempted AOD measurement at the expected time. Wireless signal strength data were used to identify units with relatively poor connection and move them into areas with better signal. In the sample deployment detailed here, no interventions based on quality control data were warranted. However, in general, these data can be used to remotely identify and address malfunctioning units mid-sample. This feature represents a substantial improvement compared

with AMODv1, which provided no sample quality control data in real time, requiring manual data acquisition (via micro SD card) and unit inspection following a failed sample.

### 3.2 Co-located reference validation

       Here, we present results of co-located validation studies for the AOD measurement system. Our cyclone-based gravimetric $PM_{2.5}$ sampling system has been validated extensively in prior work, and shown to agree closely

with reference $PM_{2.5}$ monitors (Volckens et al., 2017; Arku et al., 2018; Kelleher et al., 2018; Pillarisetti et al., 2019;





Wendt et al., 2019). Plantower light scattering sensors have likewise been evaluated extensively in prior work (Kelly et al., 2017; Zheng et al., 2018; Levy Zamora et al., 2019; Sayahi et al., 2019; Wendt et al., 2019; Bulot et al., 2019; Tryner et al., 2020).

We observed close AOD agreement between AMODv2 and AERONET monitors. Correlation plots are provided in Fig. 4 (n = 426 paired measurements per wavelength). Summary statistics are provided for each wavelength in Table 1. A plot of AMODv2 vs. AERONET co-located measurements is provided in Fig. 4.

**Table 1: Summary statistics for AMODv2 vs. AERONET co-located tests**

| Wavelength (nm) | Mean absolute error (AOD) | Deming slope coefficient | $R^2$ | AOD Precision (AOD) |
|---|---|---|---|---|
| 440 | 0.046 | 0.953 | 0.987 | 0.023 |
| 500 | 0.057 | 0.985 | 0.978 | 0.027 |
| 675 | 0.026 | 1.011 | 0.995 | 0.0089 |
| 870 | 0.033 | 1.015 | 0.977 | 0.017 |

The mean absolute errors were 0.046, 0.057, 0.026, and 0.033 AOD units at 440 nm, 500 nm, 675 nm, and 870 nm, respectively. The Deming regression slope coefficients were 0.953, 0.985, 1.011 and 1.015 at 440 nm, 500 nm, 675 nm, and 870 nm, respectively. The squares of Pearson correlation coefficients were 0.987, 0.978, 0.995, and 0.977 at 440 nm, 500 nm, 675 nm, and 870 nm, respectively (Fig. 4). With respect to precision, the average differences from the mean for units measuring coincidentally (i.e., the average amount an individual unit deviated from the mean of all units measuring at the same time) were 0.023, 0.027, 0.0089, and 0.017 AOD units at 440 nm,
500 nm, 675 nm, and 870 nm, respectively.

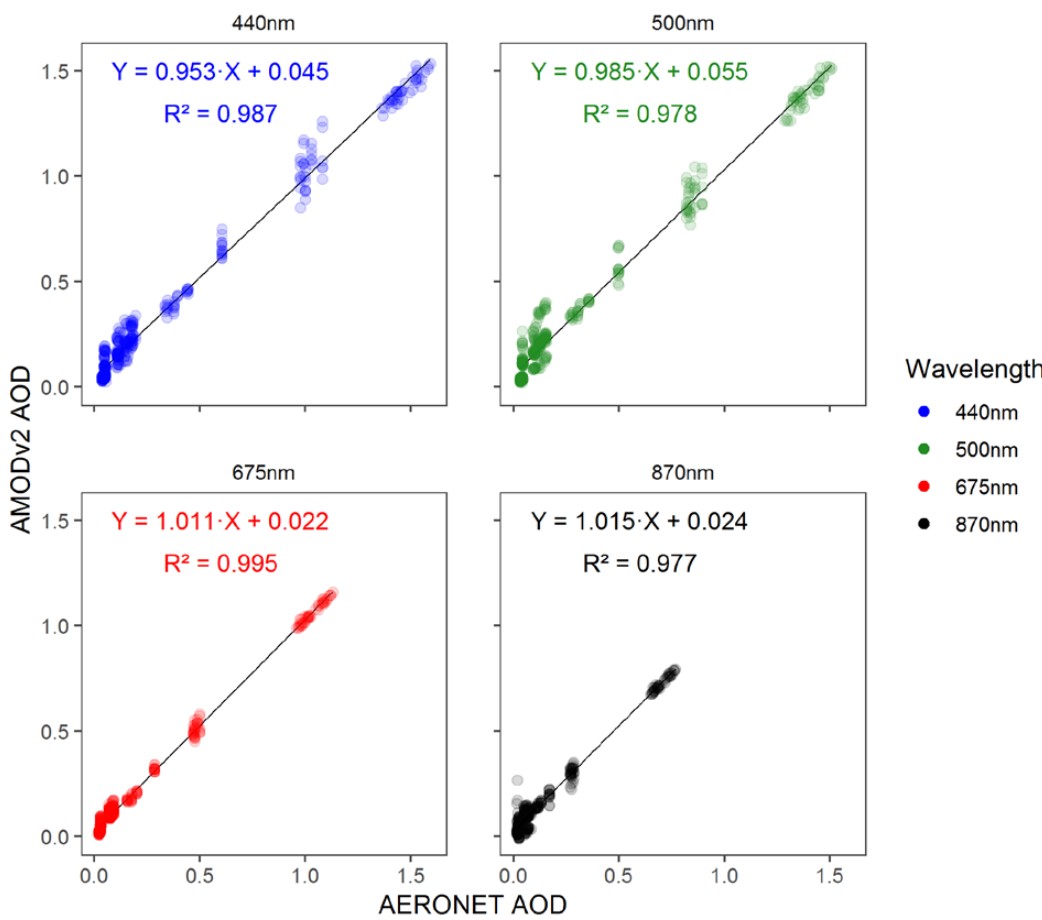

**Figure 4: AERONET (MAXAR-FUTON site in Fort Lupton, CO) vs. AMODv2 AOD co-located comparison (n=426)**
**results with panels separated by wavelength. Lines of best fit were calculated via deming regression analysis.**


   Due to the broad range of AOD levels during testing, global summary statistics do not fully capture how
error and precision scales with increasing AERONET AOD, as these figures of merit are not constant across the
range of measured AOD values. Measurements at high AOD impact the mean absolute error disproportionately,
while measurements at low AOD impact the mean percent error disproportionately. We derived the following
375 expected error (EE) equations to constrain the error of AMODv2 measurements relative to AERONET as a function
of AOD:

$$EE_{440} = \pm(0.08 + 0.05 \cdot AOD_{AERONET440}) \tag{4}$$





$$EE_{500} = \pm(0.090 + 0.040 \cdot AOD_{AERONET500}) \tag{5}$$

$$EE_{675} = \pm(0.045 + 0.020 \cdot AOD_{AERONET675}) \tag{6}$$

$$EE_{870} = \pm(0.050 + 0.010 \cdot AOD_{AERONET870}) \tag{7}$$

The bounds defined by Eqs. (4) through (7) contain 85% of the co-located measurement pairs. A

logarithmic plot illustrating how the error bounds scale with increasing AOD is provided in Fig. 5.

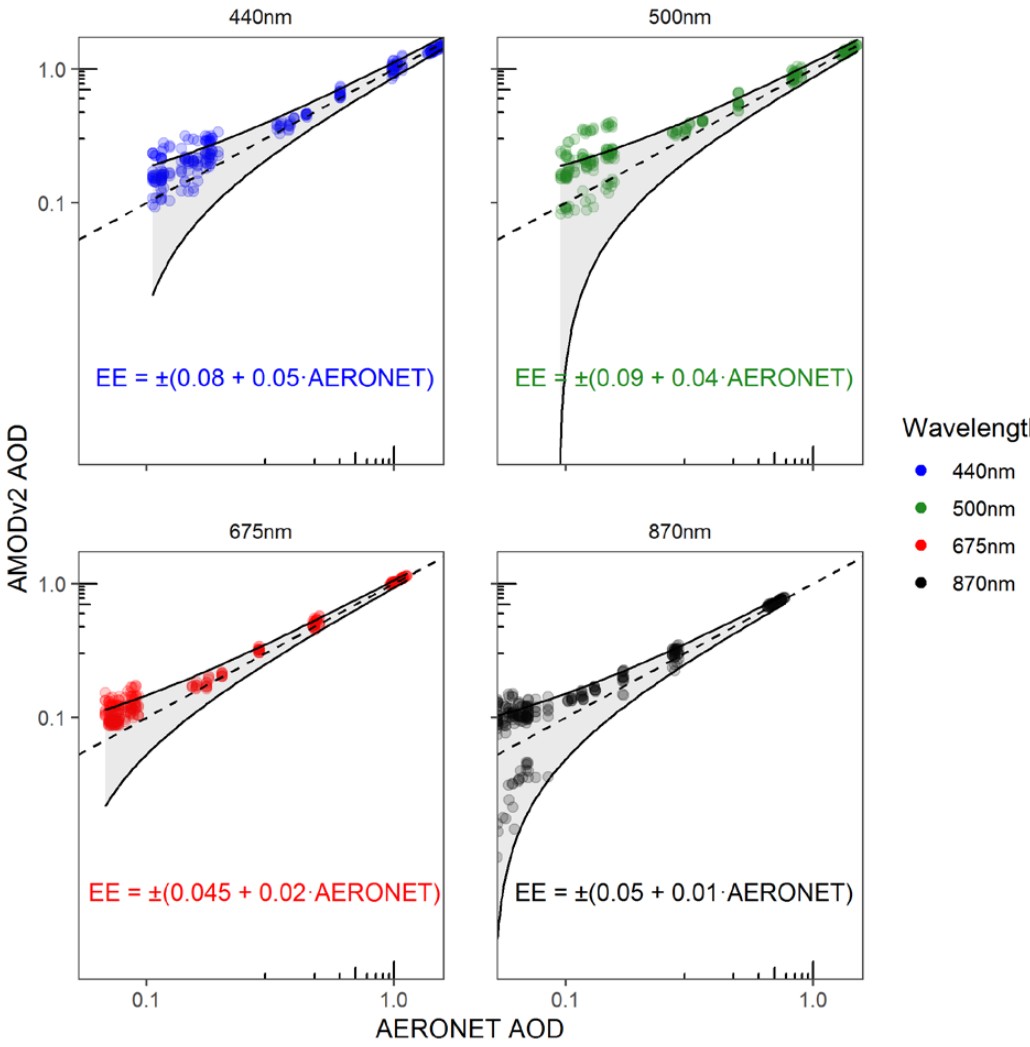

**Figure 5: Logarithmic AERONET vs. AMODv2 AOD co-located results with expected error (EE; AOD units) bounds,**

**with panels separated by wavelength. Equation bounds contain 85% of co-located measurements. The dashed line is a 1:1**

**line.**





Equations (4) through (7) indicate a low dependence of the AOD magnitude on the AMODv2 error relative to AERONET for all wavelengths. Existing error between AMODv2 and AERONET measurements was explained

primarily by the constant term.

AMODv2 bias relative to AERONET was primarily dependent on the specific unit, rather than systemic design uncertainty. A mean-difference plot colored by AMODv2 unit ID is provided in Fig. 6.

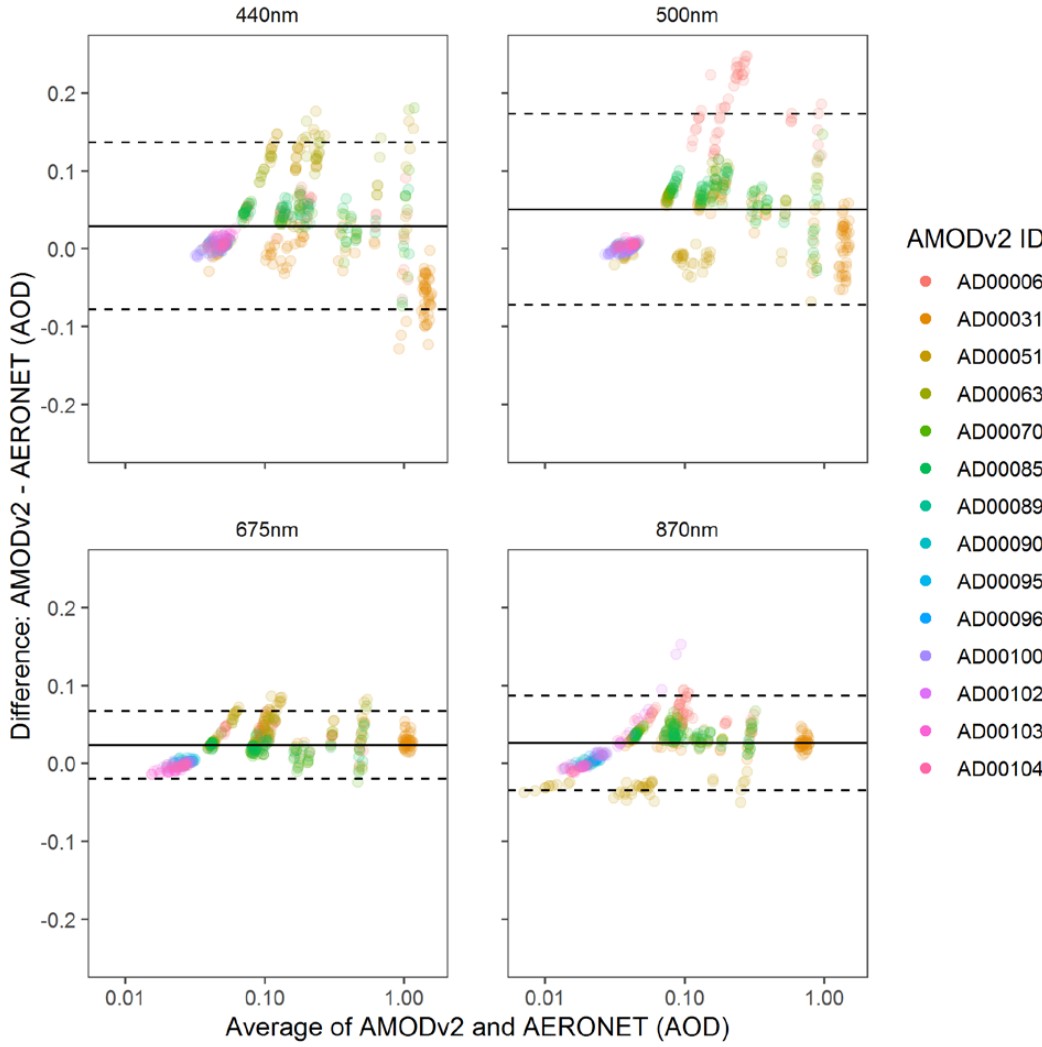

**Figure 6: Mean-difference plot for measurements taken by AERONET and AMODv2 monitors, with panels separated by**

**wavelength by wavelength. Points represent paired AMODv2 and AERONET measurements with the average of the measurement pair on the x-axis in log scale and the difference on the y-axis. The top and bottom dashed lines represent the upper and lower limits of agreement, respectively, evaluated at 95% confidence. The solid line in between the limits of agreement is the mean difference between the two measurement techniques. Points are colored according to the AMODv2 unit ID.**





Units AD00006 and AD00051 exhibited the highest bias at 440 nm and 500 nm respectively. With units
AD00051 and AD00006 removed from the data set, mean absolute errors were reduced by 0.011, 0.013, 0.0075, and
0.0043 AOD units at 440 nm, 500 nm, 675 nm, and 870 nm, respectively. Bias from units AD00006 and AD00051
also impacted the EE derivations. With units AD00006 and AD00051 omitted, Eqs. (4) through (7) bound 92.5%,
94.6%, 97.6% and 92.2% of the co-located pairs, respectively. Individual unit bias was most likely caused by faulty
calibration or optical sensor drift over time. Previous work has noted the tendency for optical interference filters to
degrade over time, changing the accuracy of the most recent calibration (Brooks and Mims, 2001; Giles et al., 2019).
Units AD00089 through AD00104 were manufactured closer to the testing dates, which may have contributed to
those units exhibiting lower measurement bias. To mitigate unit specific errors, we recommend re-calibrating
instruments at least one time per year. We will continue to monitor the performance of AMODv2 units over time as
part of ongoing work.

### 3.3 Discussion

          With the AMODv2 design presented here, we addressed the key shortcomings that we identified with
AMODv1 enumerated in the introduction. First, AOD quality control was addressed with motorized solar tracking
and a cloud screening protocol. AMODv2 AOD measurements are taken as triplets, facilitating the application of
screening protocols based on temporal variation (Smirnov et al., 2000; Giles et al., 2019). The availability of full
data files at the end of each sample facilitates additional screening based on hourly and daily variations in AOD
values, beyond the immediate quality controls applied to triplets. Second, insufficient temporal resolution was
addressed by automating AOD measurement and increasing the sample rate. With automatic sampling in place, units
measure every 20 minutes of daylight for up to five days. This updated protocol increases the likelihood that
measurements will be available at the desired times of day (e.g. satellite overpass times). Third, we reduced the
potential for operator error by eliminating the manual alignment requirement present in the prior design via solar
tracking. Fourth, we improved data accessibility through the integration of a Wi-Fi module and a user-friendly
website interface. AMODv2 data are accessible remotely as measurements are taken, enabling real-time analysis and
data-driven decision making. These design improvements were implemented while improving portability and
maintaining low cost. Unlike the AMODv1, the AMODv2 does not require a tripod mount, and can be easily
relocated without requiring manual re-alignment with the sun. The AMODv2 also samples continuously for 5 days
(3 more than AMODv1) on battery power alone. The assembly cost of one AMODv2 is 1/40th the purchase price of
an AERONET CE318 sun photometer. One AMODv2 is less than 1/12[th] the combined purchase price of commonly
used gravimetric sampling components including an inlet (PM10 – Mesa Labs SSI2.5, Lakewood, CO, USA) a
cyclone (URG Corp 2161, Chapel Hill, NC, USA), a pump (Gast 86R142-P001B-N270X, Benton Harbor, MI,
USA), and a mass flow controller (Alicat MCRW-20SLPM-D/5M, Tucson, AZ, USA). The AMODv2 assembly
price is less than 1/20[th] the retail price of reference grade light scattering monitors (GRIMM EDM 180, Ainring,
Germany; TEOM™, Waltham, MA, USA).

          The design advantages of the AMODv2 can be applied toward novel and impactful applications in the field
of air pollution research. Citizen-led sampling is a promising approach to produce large-scale data sets to quantify





air pollution concentrations at spatiotemporal resolution unachievable by more-expensive reference monitors (Brooks and Mims, 2001; Boersma and de Vroom, 2006; Ford et al., 2019). With the help of citizen volunteers, researchers can deploy instruments in greater numbers, covering areas that are more representative of population distributions. The AMODv2 was designed to give citizen volunteers an effective tool to make highly impactful air

pollution measurements from their homes or workplaces. With its low manufacturing costs (Table S1), AMODv2s can be produced and distributed widely throughout a community (Ford et al., 2019). After a short training session, volunteers with a smartphone and internet connection can begin contributing AOD and $PM_{2.5}$ data to their community data set (Ford et al., 2019). The AMODv2 is novel as a citizen-science instrument for the breadth of data each sample provides. Each AMODv2 sample includes $PM_{2.5}$ and AOD measurements taken simultaneously at 20

minutes intervals, alongside a five-day integrated filter sample. Data collection that would normally require multiple instruments, is possible with a single AMODv2 unit. Community members and stakeholders may access this rich data set in real time using the companion website for faster analysis and intervention. We continue to prepare citizen-science deployments to realize the potential of the AMODv2. We have completed smaller scale pilot campaigns, where AMODv2s were used effectively by citizen volunteers. We are planning larger deployments

which will be the subject of future manuscripts.

The datasets generated by AMODv2 deployments can be used to advance understanding of air pollution and inform efforts to mitigate its impact on public health and the environment. Satellite-based $PM_{2.5}$ retrievals are a primary source of data used to assess the global impact of air pollution (Boys et al., 2014; Brauer et al., 2016; van Donkelaar et al., 2016; Li et al., 2018; van Donkelaar et al., 2019; Lu et al., 2019). With more accurate and

informative satellite-based retrievals, the impact of air pollution can be more accurately assessed, leading to more effective strategies to control emissions and exposures. As a ground monitor measuring both $PM_{2.5}$ and AOD, AMODv2 measurements can be used to constrain the uncertainty of satellite-based AOD retrievals, as well as the reliability of the conversion between an AOD value, to a ground-level $PM_{2.5}$ estimate. The spatial resolution of satellite-based $PM_{2.5}$ retrievals is on the order of kilometers (Salomonson et al., 1989; Diner et al., 1998, 2018;

Zoogman et al., 2017; Wei et al., 2019). The temporal resolution is often on the order of a full day to a week, depending on the satellite path and period (Salomonson et al., 1989; Diner et al., 1998; Zoogman et al., 2017; Diner et al., 2018; Wei et al., 2019). This leaves variation at lower spatial and temporal scales unaccounted for in the absence of ground monitors. Multiple AMODv2s deployed in a spatially dense network may be used to evaluate the degree to which air pollution varies within the spatial resolution limits of satellite-based retrievals. The relatively

high measurement frequency of the AMODv2 ensures a low temporal discrepancy (<10 minutes) between an AMODv2 measurement and a satellite overpass. The remaining AMODv2 data can be used to assess deviations in air pollution between satellite-based retrievals. Information gained through these analyses can be used toward improving the usefulness of satellite measurements for determining surface air quality, upon which many impact assessments and mitigation strategies rely.

Our sample testing has revealed several areas of potential improvement for the AMODv2 design. Individual unit bias was the primary source of error relative to AERONET and relative to other AMODv2 units. We believe instances of individual unit bias highlights potential limitations of the existing calibration protocol. In this





work, calibration constants were determined based on co-located AMODv2 and AERONET data from a single day. We plan to explore calibrating over multiple days to include more diverse conditions in the calibration data set, and

reduce the influence of potential outlier points on the overall calibration. This approach could also highlight inconsistencies from one calibration to the next (e.g. poorly cleaned AOD lens; intermittent this cirrus), leading to more consistent device performance. Additionally, long-term durability testing of the AMODv2 has been limited. Compared with the AMODv1 design, the AMODv2 has more moving parts and is therefore more difficult to mechanically seal. One intended application of the AMODv2 is long-term (on the order of months to years) citizen

science deployments. The AMODv2 has been robust to short-term weather events (e.g. rain, wind, and snow), but we do not yet have satisfactory data on the durability of AMODv2 units in the field for longer periods of time. We will continue to monitor the performance of calibrations and mechanical robustness of AMODv2 units deployed in upcoming field campaigns.

**Conclusions**

The AMODv2 is a low-cost, user-friendly, and high-performance instrument for $PM_{2.5}$ and AOD measurements. Here, we present improvements made to the AOD measurement system and the implementation of wireless data transfer and real-time visualization, which were the primary areas of improvement compared with a previous design. We evaluated the AMODv2 under a wide range of atmospheric pollution levels and observed close agreement between the AMODv2 and AERONET AOD measurements, with mean absolute errors of 0.046, 0.057,

0.026, and 0.033 AOD units at 440 nm, 500 nm, 675 nm, and 870 nm, respectively. The portability, performance, and low cost of the AMODv2 make it a practical option to establish spatially-dense $PM_{2.5}$ and AOD monitoring networks. Such networks could provide valuable information toward the advancement of satellite remote sensing technologies and applications; as well as our understanding of how air pollution impacts public health and the environment.

*Data availability.*

All AMODv2 data collected and used in this study are available at the following URL:
https://hdl.handle.net/10217/225291

*Supplement.*

The supplement related to this article is available online at:

*Author contributions.*

JV, JRP, SJ, ML, and BF designed the study and concept for which the AMODv2 was designed. EW, CQ, DML, CL, and JV designed the AMODv2 instrument. EW, CL, and JM manufactured units. EW, BF, MC, ZR, CL, and JM designed validation experiments and analyzed validation data. CQ designed the mobile application. DHH



developed the companion website. ZR, BF, ML, and EW wrote the user manual. EW led the paper with BF, JRP, SJ, and JV; and all co-authors contributed to interpretation of results and paper editing.

*Competing Interests.*

The authors declare that they have no conflict of interest.

*Acknowledgements.*

The authors wish to thank Mollie Phillips, Nick Trammel-Jamison, and Todd Hochwitz (Zebulon Solutions LLC, Longmont, CO, USA) for their contributions to this work. The authors also thank Michele Kuester of Digital Globe and Janae Csavina of NEON for their help securing AERONET co-location sites.

*Financial support.*

This research was supported by NASA grant 80NSSC18M0120.

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
