# Peer review of "A low-cost monitor for simultaneous measurement of fine particulate matter and aerosol optical depth – Part 3: Automation and design improvements"

_Atmospheric Measurement Techniques, 2021_

## Referee Comment (RC1)

**1. Original Submission**

**1.1. Recommendation**

Major revision

**2. General comments:**

amt-2021-73

*"A low-cost monitor for simultaneous measurement of fine particulate matter and aerosol optical depth – Part 3: Automation and design improvements"*

**Overall opinion:** This is the third study submitted to AMT by these authors that is aimed to introduce and further develop AMOD sensors of PM2.5 and AOD. The first study from 2019 has present development of AMOD whereas good agreement between AMOD and collocated AERONET instruments was reported (10%), thus confirming the capability of AMOD to quantify PM2.5 and AOD. The second study showed promising results from a pilot field campaign (2017) in Colorado where several AMOD sensors were united in the network. AMOD sensors were able to provide spatial variability of AOD and PM2.5 at fine scales (compared to satellites and PM2.5 models). The current study (#3) presents AMODv2 (autonomous version of AMOD) that is able to measure PM2.5 and AOD with 20-minute time interval. Also, the authors show the ability of AMODv2 to provide AOD and PM2.5 during a wildfire smoke event and prove that AMODv2 results have high agreement with AERONET observations. While I acknowledge the undoubtful scientific importance of the works #1 and #2, the current study looks significantly weaker from scientific perspective. The improvement of spatio-temporal sampling of sensors and technical improvements are important per se, but unlikely deserves a standalone manuscript. The current efforts of the authors are interesting, but they do not follow the recommendations given in the articles #1 and #2. More specifically, #1 and #2 had encouraged to (a) investigate sensitivity and stability of sensors in variable environmental conditions and under different weather conditions (#1), to (b) make more comprehensive investigation of local air quality and (c) provide the information that can be used in conjunction with the satellites (b,c come from #2). It wouldn't be a problem for a random article on that topic as each research work spawns several independent vectors, but I do see almost the same roster of the authors in this study and it is called "Part 3".

Hence, I have an impression that the authors underachieved their own long-term scientific goals and submitted the effort with the engineering improvement of their sensors (better temporal resolution, wireless connection, accessibility, website) without sound research findings. At one hand, the previous two works provide a strong scientific and reputational basis for pushing the current article to publication. At other hand, many aspects of these sensors had been discussed by the authors in #1 and #2 works and therefore every new work requires the higher bar for providing useful scientific implications, not just reporting the technical improvement of their sensors and subtly advertising them in between. I give more concrete arguments supporting my viewpoint in the minor comments section.

The best chance to increase the scientific value of this article for the authors is to follow their own past recommendation in a way the current research design allows it. From my perspective, it has sense, first, to analyze systematic bias, instrumental errors and instability of the sensors vs AERONET in clean air conditions and during the biomass burning event. Only then, to provide

Figure 3-alike plots with biomass burning event and clear conditions whereas AOD estimates within specific time interval should be corresponded with uncertainties calculated in the former part of the analysis. During their analysis, the authors should clearly distinguish the technical improvements of sensors that are not directly related to atmospheric measurements because they are not quantified here (website access, wireless connection etc) and the improvements of the actual atmospheric techniques that are **quantitatively evaluated** and supported by their findings. I believe that the only the latter aspect (atmospheric-related improvements) is important for the journals as AMT, while the simple technical improvements can be just summarized and present in a table.

**2.1. Specific comments:**

1. **Abstract.** There are not enough details about how the sensors performed under wildfire smoke event. Was it very different from clean conditions? Was the systematic bias affected by the presence of biomass burning particles in the air or their concentration? What we can learn from this except the fact that AMODv2 did work and it's a good job of the engineers? Also, I think the implications about evaluating remote sensing observations and atmospheric modelling are very ambitious, but hurriedly formulated for too broad scales. What kind of remote sensing instruments that measure AOD the authors are going to evaluate with this data, all of them? The sensors are ground-based instruments, while some aerosol remote sensing instruments (especially the active ones) are advantageous because they allow retrieving aerosol microphysics from various heights. If the remote sensing instrument has a scanning capability, then maybe hundreds of sensors mounted on various heights of multiple high towers densely scattered around the observation area can help evaluating aerosol optical or microphysical properties from this instrument. I think the authors understood my idea, without specifying the type of remote sensing instrument (and model as well), these implications are doubtful and an easy target for criticism.

2. **Research Aim and Objectives**. "The primary objective of this current work was to design and integrate a system for automatic multi-wavelength AOD measurements throughout daylight hours and to validate the performance of this system against AERONET". Several critical comments here. (1) only AOD is mentioned, what about PM2.5? (2) the authors mentioned that they showed stability of the sensors in wildfire smoke event (see the abstract). This point is scientifically valuable, but unfortunately it was not even a research objective in this article. This decreases scientific values of the findings narrowing them to technical improvement of a specific instrument and confirming my concerns from above. (3) The authors mention the limitations for AMOD including problem with quality control measures, cloud contamination and misalignment, but it is not clearly articulated in the research aim whether the authors were going to solve all these problems (and therefore these are research objectives) or just listed them to confine their research domain.

3. **Format.** I encourage the authors to check the consistency of precision of reported AOD estimates. For instance, in line 40 AOD is reported with 3 digits after zero and then with two digits after zero. Actually, this choice should depend on the actual accuracy of the instrumentation and be consistent and realistic. In line 320 AOD is provided with 1 digit after zero, same for line 324. Figures have poor quality, I think. Figure 3 is hard to interpret due to color choice for instance. Also, I don't know if it is related to the journal policy or not, but only every 5[th] line is marked in the submitted PDF. It's more convenient for both the authors and the reviewers, the PDF with every line marked.

4. **Accuracy, stability and instrumental errors**. The uncertainty estimates are missing for AOD and PM2.5 (abstract) seemingly due to fragmental lack of the information about accuracy, systematic noise and stability of these sensors throughout the article. Table 1 presents such information (without highlighting whether this analysis has been provided for clear or biomass burning or mixed conditions). Despite this, Figure 3 lacks errorbars. The problem of some cheap nodes for registering PM2.5 is that they exhibit extremely strong and unrealistic temporal variations (by magnitude) due to presence of smoke or dust (high AOD). Much of these observations usually have too low signal-to-noise ratio. Lines 315-328 paragraph reports AOD and PM2.5 statistics of a severe pollution event whereas no uncertainties are reported. It is hard to believe that the magnitude of uncertainty has not been affected at PM2.5 levels of $>250$ μg m$^{-3}$. This is critical, I think. The authors should use their own findings from this article (Table 1) about this or provide adequate information from their previous works.

**2.2. Minor comments:**

5. Lines 34-36. It's my subjective opinion, but the way how this sentence is formulated sounds more like a commercial subtle advertisement. After re-reading the abstract, I found out that there are only couple of sentences about user applications of AMODv2, but the way how the authors formulated it convinced my mind that 50% of the information from the abstract was similar to a flyer of commercial sensor.
6. Line 50. I know the aerosol climate effects, but what is role of aerosols in "environmental change" worldwide requires additional explanation.
7. Line 67 "Specialized equipment", I think the authors can be more concrete and say "aerosol remote sensing instrumentation"
8. Line 68. AOD is a direct measure for atmospheric extinction, it is not just related.
9. Lines 99-100. I think when one is discussing the deployment of simple AOD (or PM2.5) sensors, the common concern was a tradeoff between their cheapness (and possibility to create a very dense sampling network) and their notorious instability especially in high relative humidity conditions or just variable weather conditions. When use of low-cost sensor network is proposed, one should always mention about their accuracy or the tradeoff between accuracy and easiness/cheapness/simplicity.
10. Line 109. More information about also systematic noise, instrumental error and sensors stability (all this information determined in work #1 or #2) should be explicitly provided here. This is a problem for the entire article as it supposed to be a strong point, but actually ends up as a weak one.
11. Line 133. What about accuracy of a sensor? This attribute is implied in "mechanical robustness"?
12. Line 266-267 Once again I have a feeling of advertisement.
13. Line 279. Is temporal variability-driven uncertainty of average AOD (PM2.5) reported for a user or for a reader of this article?
14. Line 288. It would be nice to have information about the distances of these sensors between each other. They are collocated? It is confusing because the authors say "we collocated our instruments within 50 m" but it is not clear whether AERONET-sensor couples were collocated (I guess they don't have 10 sun photometers) or sensor-to-sensor collocation was made based on 50 m distance between them. This collocation criteria should be more clearly articulated; some table of sensor locations or map would be useful otherwise.
15. Line 301. Any reference to these wildfires? What is the argument led to conclusion that the instrument was measuring exactly smoke during their analysis?
16. Line 307. I think the statement about accuracy of AMODv2 is hasty without providing details about their stability.

17. Figure 3, are these realistic spikes in PM2.5? 300 μg/m$^{-3}$. What is the uncertainty of this spike? 30 μg/m$^{-3}$ or more? How realistic are temporally averaged estimates of PM2.5 during such peaky periods of high aerosol load? Is such strong temporal variability reflected in some parameter for a user of this sensor or (more importantly) for a reader of this scientific article? I cannot distinguish most points; I see mostly magenta points. Why the authors use so visually inconvenient color scale? There are no black or gray points but I see two types of magenta points that are undistinguishable. All points have kind of pastel tones making it even worse, blue looks like red, red looks like magenta, etc.

18. Line 319. Are there any other arguments confirming the "moderate air pollution event" in the area except AMODv2 that is being tested and evaluated in this study?

19. Line 329. Once again, the website is mentioned, what does it give to a reader that data from the sample deployment were accessed from this website? Redundant information.

20. Line 332. Statement about improvement of AMODv2 wireless connection. There should be some kind of table whereas technical characteristics (and superiority) of AMODv2 that are not related to atmospheric measurement techniques are directly shown versus inferior AMODv1.

21. Line 333. "Weatherproof design" The authors should either provide arguments for extensive evaluation of these sensors in variable weather conditions proving that sensors are actually weatherproof or move this description for the aforementioned table. In the first case, a test for low and high temperatures, variable humidity conditions and mixed aerosol conditions should be shown. Several geographic locations are desirable as we cannot conclude about such stability just based on observations in Colorado. In the case if the authors decide to make weatherproof evaluation, I foremost recommend to check whether their PM2.5 concentrations are subjected to influence of high relative humidity as shown in Crilley et al., (2020) whereas the authors have noticed hygroscopic growth of aerosols under >60% RH conditions.

22. Line 335. Once again, some redundant details of technical improvements of sensors in the journal dedicated to atmospheric measurement techniques. "Data accessibility" can be also moved to the table with comparison of AMODv1 and AMODv2. Also, the description of this paragraph is purely qualitative. If some engineers want to check the key indicators of stability of these sensors, they cannot because there are no such quantitative data here. If atmospheric scientists read this, once again, redundant qualitative information for the main text of the journal like AMT.

23. Lines 354-356. Only biomass burning event is analyzed here or also clean conditions included?

24. Table 1. AOD precision and mean absolute error are finally provided here. Systematic uncertainties should be reported when AOD estimates are reported throughout the manuscript.

25. Line 360. What about mean absolute errors depending on the environmental conditions, they varied from biomass burning event and clean conditions?

26. Figure 4. "Fort Lupton, CO" I think it's better to explain CO as Colorado if mentioned for non-American readers.

27. Line 389 "was explained primarily by the constant term" that is one of the reasons why uncertainties should be explicitly shown.

28. Line 391 Once again it's unclear whether only biomass burning event or mixed events are implied?

29. Line 404. and again, instrumental errors due to optical sensor drift over time should be timely quantified in the manuscript to be used for AOD estimates the authors report for analyzing biomass burning event

30. Line 407. How large are discrepancies in the production dates between these sensors actually to result to such differences?

31. Line 411. Denoting discussion as a subsection is rather uncommon.

32. Lines 420-433. Technical improvements such as data accessibility, protocol, time resolution of observations without indications how it can improve the agreement with the referenced AOD/PM2.5 observations or fill the existing gaps in this field are really minor contribution for atmospheric sciences given the existence of works #1 and #2 on this topic.
33. Line 427. Price of a sensor is an important information but given the lack of really useful conclusions for atmospheric science, the price reference just looks odd in discussion.
34. Line 434-439 Other types of cheap AOD sensors can be also (and already being) used for this purpose. The authors should nail down the implications for citizen science provided particularly by improvement version of AMOD (v2) since the articles about AMODv1 were already present.
35. Line 445 "Data collection that would normally require multiple instruments is possible with a single AMODv2 unit" this sentence requires explanation
36. Line 446. Once again, there are other instruments and also cheap sensors that have been already exploited for this purpose (with less than a minute temporal sampling of PM2.5), so what's new here?
37. Line 479. Proposal to use AMODv2 in citizen science is repetitive
38. Line 484. The conclusions look like discussion about the scientific findings of this article that should be extended. Meanwhile, the two latest paragraphs of the discussion look like conclusions of this article, which are however, not based on the findings about evident advantages of AMODv2 for atmospheric science, but on minor technical improvements of this type of sensor.
39. Line 490. The proposal about establishing network for AOD monitoring is repetitive given the information provided in the discussion. The authors are advised to thoroughly check which information they will use as discussion and which as conclusion. This should be done for avoiding logical loops and repetitions.

---

## Referee Comment (RC2)

**A low-cost monitor for simultaneous measurement of fine particulate matter and aerosol optical depth – Part 3: Automation and design improvements**

The authors present improvements in AMOD sensing device for concurrent PM2.5 and AOD measurements. They describe a sample deployment of 10 AMODv2s for 2.5 days during a wildfire smoke event, and they present a 7-day validation campaign of the AMODv2 that was co-located with a AOD reference device.  The results indicate close agreement between AMODv2s and the reference instrument  although they identified some intra-device variability. They conclude that the AMODv2 is well suited for citizen science and other high-spatial-density deployments due to its low cost, compact form, user-friendly interface, and high measurement frequency of AOD and PM2.5.

The paper addresses an important topic, a cost-effective strategy for improving satellite estimates of PM2.5 concentration.  They present results to support their statements about measurement accuracy.  However, it presents rather limited data to support many of the authors' other conclusions.  The paper would be strengthened with additional data.  It also requires some clarification regarding real-time presentation of PM2.5 concentrations as well as some date inconsistencies.  If additional results are not possible, the manuscript may be more suitable as a technical note.

**Significant weaknesses**

*Limited scope of data.* The website presented in the paper illustrates that the AMODv2s are collecting and presenting measurements to the public, which is great.  It is puzzling why the authors present only 2.5 days of measurements (2 days of AOD measurements) from 14 devices. This limited data set also makes it difficult to support the authors statements that the system is autonomous.  The paper would be greatly strengthened if they:
- Presented longer-term measurements, which is one of the key motivations for this developing autonomous measurements
- Co-located a few AMODv2s to understand intra-device variability, and have long-term validation measurements by co-locating the AMODv2s with AOD reference devices for a long time period (seasons to a year).  The manuscript provides some evidence that device to device variability could be important.
- Presented data that indicates the autonomous nature of the device, % of device up time, and number of valid AOD measurements (quality control described with triplicate measurements), number of AOD measurements excluded for cloud cover, etc.   This needs to be done for weeks or months of measurements to begin to support the autonomous claims in the manuscript.

The validation measurements of AMODv2 are also limited.  The authors state that their validation measurements cover a range of conditions, which is important.  If the AMODv2 is capable of autonomous measurement, it is unclear why the validation results are so limited.  It is also unclear how the authors came to 426 data paired points.  I would encourage them to include a table in the supplementary materials that presents the device ID, the number of valid AOD measurements each day, the number and % of excluded measurements (if any), % of missing data and the cloud and PM2.5 conditions of the day.  I struggled with understanding how 426

paired data points were achieved. If the sampling rate was 2.5 to 3 minutes, are they presenting the results of each test (one every two to three hours, this would get closer to 426 data points were the samples averaged? ) or samples selectively included.

With less than 10 days of data total (2.5 of sample deployment and 7 days of validation), it is difficult to see how this annual calibration recommendation is reached.   Line 408: "To mitigate unit specific errors, we recommend re-calibrating instruments at least one time per year."

*PM2.5 concentrations.* The paper presents PM2.5 measurements from the Plantower PMS5003. The authors need to clarify if they are presenting measurements that were corrected with their on-board filter measurements (or if these are raw measurements or corrected using some other strategy).  They should provide these filter-based correction factors in the supplementary material.  If they are using some other method, they should describe this. They should also describe if they are applying corrections to Plantower's raw ATM or CF=1 measurements.

Also, in several places in the manuscript, the authors discuss real-time visualization and data access.  However, as the authors and others have published previously the Plantower PMS 5003 raw measurements are biased.  They require a correction factor for the aerosol conditions of interest.  Reasonably accurate real-time PM2.5 concentration estimates would not be possible if the filter cartridges require one week of sampling, must be mailed in, and weighed.  These corrections would only be possible weeks after the PM2.5 measurements are posted. The authors need to clarify what they are presenting and if the measurements are corrected how they are corrected.

**Moderate weaknesses**
Table 1.  What types of conditions are being presented here.  All 7 days or a subset?  How do the statistics vary by condition?

The manuscript in places reads a little like an advertisement for the AMODv2.  The authors should focus on statements that are supported by their results (i.e., the terms autonomous, real-time, user friendly, use for citizen science).  See comments about autonomous and real-time in the previous section.

Figure S7 does not occur during either the sample deployment (October 2021 or maybe 2020, see comment on date inconsistencies below) or during the co-location validation testing.  It is unclear why the authors would select a time other than the sample deployment described in the paper.  If there is a good reason, they should provide it.  They should also discuss the causes of the PM2.5 concentration peaks.  Also, this figure illustrates that more than one half of the AMODv2s are offline (see comment about autonomous).

Given that the AMODv2 requires WiFi, wouldn't it be reasonable to have a plug in option, which would likely increase up-time, see comment regarding figure S7.

It is unclear why the authors present only 2.5 days of results when the devices are capable of 5 days of measurements.

This may be just my eyes, but Figure 4 doesn't look like it contains 426 data points.

The Figure 2 shows one sampling interval rather than continuous sampling. The filter cartridge needs to be replaced and the instrument restarted. This should be included in the figure.

Line 288 The authors discuss their sample deployment of co-located 10 AMOD units during a wildfire smoke event in Fort Collins, Colorado in October of 2021. I believe that they mean October 2020 since October 2021 has not yet occurred.

Line 374 The authors should clarify how they derived their expected error equations.

Line 427. The authors are not comparing apples to apples here. First, they are comparing their component costs to commercial instruments, which are likely 5 to 10X the component costs. Further the GRIMM EDM is a Federal Reference Method. A more reasonable comparison would be a low-cost particulate matter sensor, i.e., Purple Air at $250 or something like the TSI DustTrak $5000 with a cyclone cut point and filters that can be weighed.

**Minor**
Line 395 "by wavelength" is repeated

---

## Author Comment (AC1)

We thank the reviewers for their insightful comments, which have allowed us to produce a stronger manuscript. Our responses to the general and specific comments are given below. Reviewer comments are provided in italics and our responses are given in plain text. Line number references pertain to the revised manuscript with no tracked changes. Quoted text from the revised manuscript is given in blue in this response document.

**Reviewer 1:**

Reviewer 1 general comments

*Overall opinion: This is the third study submitted to AMT by these authors that is aimed to introduce and further develop AMOD sensors of PM2.5 and AOD. The first study from 2019 has present development of AMOD whereas good agreement between AMOD and collocated AERONET instruments was reported (10%), thus confirming the capability of AMOD to quantify PM2.5 and AOD. The second study showed promising results from a pilot field campaign (2017) in Colorado where several AMOD sensors were united in the network. AMOD sensors were able to provide spatial variability of AOD and PM2.5 at fine scales (compared to satellites and PM2.5 models). The current study (#3) presents AMODv2 (autonomous version of AMOD) that is able to measure PM2.5 and AOD with 20-minute time interval. Also, the authors show the ability of AMODv2 to provide AOD and PM2.5 during a wildfire smoke event and prove that AMODv2 results have high agreement with AERONET observations. While I acknowledge the undoubtful scientific importance of the works #1 and #2, the current study looks significantly weaker from scientific perspective. The improvement of spatio-temporal sampling of sensors and technical improvements are important per se, but unlikely deserves a standalone manuscript. The current efforts of the authors are interesting, but they do not follow the recommendations given in the articles #1 and #2. More specifically, #1 and #2 had encouraged to (a) investigate sensitivity and stability of sensors in variable environmental conditions and under different weather conditions (#1), to (b) make more comprehensive investigation of local air quality and (c) provide the information that can be used in conjunction with the satellites (b,c come from #2). It wouldn't be a problem for a random article on that topic as each research work spawns several independent vectors, but I do see almost the same roster of the authors in this study and it is called "Part 3".*

*Hence, I have an impression that the authors underachieved their own long-term scientific goals and submitted the effort with the engineering improvement of their sensors (better temporal resolution, wireless connection, accessibility, website) without sound research findings. At one hand, the previous two works provide a strong scientific and reputational basis for pushing the current article to publication. At other hand, many aspects of these sensors had been discussed by the authors in #1 and #2 works and therefore every new work requires the higher bar for providing useful scientific implications, not just reporting the technical improvement of their*

*sensors and subtly advertising them in between. I give more concrete arguments supporting my viewpoint in the minor comments section.*

*The best chance to increase the scientific value of this article for the authors is to follow their own past recommendation in a way the current research design allows it. From my perspective, it has sense, first, to analyze systematic bias, instrumental errors and instability of the sensors vs AERONET in clean air conditions and during the biomass burning event. Only then, to provide Figure 3-alike plots with biomass burning event and clear conditions whereas AOD estimates within specific time interval should be corresponded with uncertainties calculated in the former part of the analysis. During their analysis, the authors should clearly distinguish the technical improvements of sensors that are not directly related to atmospheric measurements because they are not quantified here (website access, wireless connection etc) and the improvements of the actual atmospheric techniques that are quantitatively evaluated and supported by their findings. I believe that the only the latter aspect (atmospheric-related improvements) is important for the journals as AMT, while the simple technical improvements can be just summarized and present in a table.*

**Response:** We agree with Reviewer 1 that the scientific aims and inquiries that we highlight in our earlier manuscripts (Wendt et al., 2019; Ford et al., 2019) are worthy of investigation. Indeed, our team is working towards those scientific goals and we expect near-term publications that describe ongoing deployments and data integration with satellite remote sensing (note: these science-focused manuscripts fall outside of the scope of AMT and will be published elsewhere). However, we respectfully disagree with Reviewer 1 that the technological achievements reported here are unworthy of a standalone manuscript. We believe that automation of a low-cost, multi-wavelength AOD instrument (along with wireless data transmission capability) represent a significant technological achievement. We feel that these technological advances (and their evaluation/validation) align well to the scope of *Atmospheric Measurement Techniques* (as stated on the journal homepage). We also believe our investigation of the stability of AMODv2 AOD over a broader range of air pollution levels addresses the recommendation of Part 1 of this series. We hope that the reorganization of our manuscript, along with additional data/analyses as requested by the reviewers will convince them of the like.

Reviewer 1 specific comments

Major comments

1. **Comment:** *Abstract. There are not enough details about how the sensors performed under wildfire smoke event. Was it very different from clean conditions? Was the systematic bias affected by the presence of biomass burning particles in the air or their concentration? What we can learn from this except the fact that AMODv2 did work and it's a good job of the engineers?*

*Also, I think the implications about evaluating remote sensing observations and atmospheric modelling are very ambitious, but hurriedly formulated for too broad scales. What kind of remote sensing instruments that measure AOD the authors are going to evaluate with this data, all of them? The sensors are ground-based instruments, while some aerosol remote sensing instruments (especially the active ones) are advantageous because they allow retrieving aerosol microphysics from various heights. If the remote sensing instrument has a scanning capability, then maybe hundreds of sensors mounted on various heights of multiple high towers densely scattered around the observation area can help evaluating aerosol optical or microphysical properties from this instrument. I think the authors understood my idea, without specifying the type of remote sensing instrument (and model as well), these implications are doubtful and an easy target for criticism.*

**Response:** We have added details regarding the stability of AMODv2 AOD measurements between clear days and days with elevated AOD in lines 36-38 as follows:

"We derived empirical relationships relating the reference AOD level with AMODv2 instrument error and found that the mean absolute error in the AMODv2 deviated by less than 0.01 AOD units between clear days and elevated-AOD days and across all wavelengths."

We have also added Table S1, which gives summary statistics for both clean days and days with elevated AOD:

**Table S1:** AMODv2 validation summary statistics calculated separately for elevated-AOD days and clear days. Elevated-AOD days were defined as days when the average AERONET AOD at 500 nm was greater than or equal to 0.15. Clear days were defined as days in which the average AOD was less than 0.15. In total, five days were identified as clear and four days were identified as elevated-AOD.

| Wavelength (nm) | AERONET 500 nm AOD magnitude | Number of samples | Mean absolute error (AOD) | AOD Precision (AOD) |
| --- | --- | --- | --- | --- |
| 440 | All included | 426 | 0.04 | 0.02 |
| 500 | All included | 426 | 0.06 | 0.02 |
| 675 | All included | 426 | 0.03 | 0.01 |
| 870 | All included | 426 | 0.03 | 0.02 |
| 440 | > 0.15 (elevated) | 115 | 0.05 | 0.02 |
| 500 | > 0.15 (elevated) | 115 | 0.05 | 0.02 |
| 675 | > 0.15 (elevated) | 115 | 0.03 | 0.01 |
| 870 | > 0.15 (elevated) | 115 | 0.03 | 0.01 |

| 440 | < 0.15 (clear) | 311 | 0.04 | 0.02 |
| 500 | < 0.15 (clear) | 311 | 0.06 | 0.03 |
| 675 | < 0.15 (clear) | 311 | 0.02 | 0.01 |
| 870 | < 0.15 (clear) | 311 | 0.03 | 0.02 |

For the second part of this comment (regarding "implications about remote sensing and modelling"), we are unclear as to what section of the manuscript the reviewer is referring to. We suspect this comment applies to the last sentence of the abstract, which we have subsequently deleted.

*2. **Comment:** Research Aim and Objectives. "The primary objective of this current work was to design and integrate a system for automatic multi-wavelength AOD measurements throughout daylight hours and to validate the performance of this system against AERONET". Several critical comments here. (1) only AOD is mentioned, what about PM2.5? (2) the authors mentioned that they showed stability of the sensors in wildfire smoke event (see the abstract). This point is scientifically valuable, but unfortunately it was not even a research objective in this article. This decreases scientific values of the findings narrowing them to technical improvement of a specific instrument and confirming my concerns from above. (3) The authors mention the limitations for AMOD including problem with quality control measures, cloud contamination and misalignment, but it is not clearly articulated in the research aim whether the authors were going to solve all these problems (and therefore these are research objectives) or just listed them to confine their research domain.*

**Response:** We have re-formulated our research objectives to more accurately describe the aims and scope of our research efforts. With respect to the specific critical comments: 1) We do not mention $PM_{2.5}$ because that aspect of the AMOD did not undergo significant design changes and the underlying technology has been validated extensively as stated on lines 355-358 as follows:

"Our cyclone-based gravimetric PM2.5 sampling system has been validated extensively in prior work and shown to agree closely with reference PM2.5 monitors (Volckens et al., 2017; Arku et al., 2018; Kelleher et al., 2018; Pillarisetti et al., 2019; Wendt et al., 2019). Plantower light scattering sensors have likewise been evaluated extensively in prior work (Kelly et al., 2017; Zheng et al., 2018; Levy Zamora et al., 2019; Sayahi et al., 2019; Wendt et al., 2019; Bulot et al., 2019; Tryner et al., 2020)."

2) We have now explicitly included extended stability analysis as one of our research objectives on lines 127-129 as follows:

"Another shortcoming of our work on AMODv1 was limited stability analysis of the AOD sensors across varying atmospheric conditions and over time. The second objective of this work, therefore, was to evaluate the stability of the AOD sensors across a range of pollution levels and to assess the stability of the AOD sensors after repeated deployments over the course of a year."

3) We have clarified that addressing the four design shortcomings was a primary research objective on lines 121-127 as follows:

"Despite the promise of the AMODv1, the initial deployment highlighted several key limitations. First, the AMODv1 lacked quality control measures for misalignment or cloud contamination during the measurement period. Second, the instrument had limited temporal resolution for AOD (typically one measurement per day). Third, despite the presence of a visual alignment aid (Wendt et al., 2019), many volunteers found it difficult to align the instrument with the sun, which was compounded by inconsistent standards as to what constituted proper alignment. Fourth, data could not be transmitted wirelessly or accessed remotely. The first objective of this current work was to address these four major limitations of the AMODv1 design."

3. **Comment:** *Format. I encourage the authors to check the consistency of precision of reported AOD estimates. For instance, in line 40 AOD is reported with 3 digits after zero and then with two digits after zero. Actually, this choice should depend on the actual accuracy of the instrumentation and be consistent and realistic. In line 320 AOD is provided with 1 digit after zero, same for line 324. Figures have poor quality, I think. Figure 3 is hard to interpret due to color choice for instance. Also, I don't know if it is related to the journal policy or not, but only every 5th line is marked in the submitted PDF. It's more convenient for both the authors and the reviewers, the PDF with every line marked.*

**Response:** We have updated reported AOD values in the main text to consistently include 2 digits after zero. With respect to figures, we follow the recommended DPI specifications for PNG figure preparation of AMT. We suspect quality issues are the result of issues embedding within MS Word and that figure quality will improve with the final typeset version (as was the case with parts 1 and 2 of this series of articles). Figure 3 has been replaced in the main manuscript by more salient analyses and is now present in the supplement with a new color palette. Line labels for every 5th line is the policy of AMT and is part of their provided manuscript template.

4. **Comment:** *Accuracy, stability and instrumental errors. The uncertainty estimates are missing for AOD and PM2.5 (abstract) seemingly due to fragmental lack of the information about accuracy, systematic noise and stability of these sensors throughout the article. Table 1 presents such information (without highlighting whether this analysis has been provided for clear or biomass burning or mixed conditions). Despite this, Figure 3 lacks errorbars. The problem of*

*some cheap nodes for registering PM2.5 is that they exhibit extremely strong and unrealistic temporal variations (by magnitude) due to presence of smoke or dust (high AOD). Much of these observations usually have too low signal-to-noise ratio. Lines 315-328 paragraph reports AOD and PM2.5 statistics of a severe pollution event whereas no uncertainties are reported. It is hard to believe that the magnitude of uncertainty has not been affected at PM2.5 levels of >250 µg m-3. This is critical, I think. The authors should use their own findings from this article (Table 1) about this or provide adequate information from their previous works.*

**Response:** We have amended the abstract to include AERONET AOD uncertainties when referencing the AOD testing range. These uncertainties are from the AERONET version 3 paper (Giles et al., 2019). Lines 32-36 have been updated as follows:

"We present the results of a nine-day AOD validation campaign where AMODv2 units were co-located with an AERONET (Aerosol Robotics Network) instrument as the reference method at AOD levels ranging from $0.02 \pm 0.01$ to $1.59 \pm 0.01$. We observed close agreement between AMODv2s and the reference instrument with mean absolute errors of 0.04, 0.06, 0.03, and 0.03 AOD units at 440 nm, 500 nm, 675 nm, and 870 nm, respectively."

The summary statistics in Table 2 are for the full set of validation data with clear days and biomass burning days all included, without separation. We have added an extended table of summary statistics to the supplementary material (Table S1). This table includes validation metrics calculated from the global data set, in addition to separate statistics for biomass burning and clear measurements.

We have also moved Section 3.1 to the supplementary material and added uncertainty values to specific $PM_{2.5}$ and AOD measurements mentioned. For $PM_{2.5}$ measurements below 20 µg m$^{-3}$, we use the uncertainty estimates derived in the AMODv1 paper. For higher $PM_{2.5}$ concentrations we apply a 40% uncertainty estimate from Barkjohn et al. (2020). For a detailed discussion on the uncertainty of the $PM_{2.5}$ measurements, see our response to comment 17.

We have restructured the manuscript to include the following sections, which we believe better support the research objectives of this manuscript:
- 3.1 Summary and table of design improvements
- 3.2 AOD sensor validation and calibration stability
- 3.3 Reliability testing

Minor Comments

*5. **Comment:** Lines 34-36. It's my subjective opinion, but the way how this sentence is formulated sounds more like a commercial subtle advertisement. After re-reading the abstract, I*

*found out that there are only couple of sentences about user applications of AMODv2, but the way how the authors formulated it convinced my mind that 50% of the information from the abstract was similar to a flyer of commercial sensor.*

**Response:** We have edited the functionality description of the AMODv2 in the abstract on lines 24-32 as follows:

"Here in part 3, we present an updated version of the AMOD, known as AMODv2, featuring design improvements and extended validation to address the limitations of the AMODv1 work. The AMODv2 measures AOD and PM2.5 at 20-minute time intervals. The sampler includes a motorized sun-tracking system alongside a set of four optically filtered photodiodes for semi-continuous, multi-wavelength (current version at 440, 500, 675, and 870 nm) AOD sampling. Also included are a Plantower PMS5003 sensor for time-resolved optical PM2.5 measurements and a pump/cyclone system for time-integrated gravimetric filter measurements of particle mass and composition. AMODv2 samples are configured using a smartphone application and sample data are made available via data streaming to a companion website (csu-ceams.com)."

6. ***Comment:*** *Line 50. I know the aerosol climate effects, but what is role of aerosols in "environmental change" worldwide requires additional explanation.*

**Response:** Here "environmental change" does refer to climate change. We have revised lines 47-50 for clarity as follows:

"Fine particulate matter air pollution (PM2.5) is a leading cause of human morbidity and mortality, and also a significant contributor to radiative climate forcing (Myhre et al., 2013; Forouzanfar et al., 2016; Brauer et al., 2016; Vohra et al., 2021)"

7. ***Comment:*** *Line 67 "Specialized equipment", I think the authors can be more concrete and say "aerosol remote sensing instrumentation"*

**Response:** Thank you for the suggestion. We have revised lines 64-67 as follows:

"Satellites equipped with aerosol remote sensing instrumentation retrieve aerosol optical depth (AOD), a measure of light extinction in the atmospheric column, which can then be converted to ground level PM2.5 using a CTM or statistical relationship."

8. ***Comment:*** *Line 68. AOD is a direct measure for atmospheric extinction, it is not just related.*

**Response:** The reviewer is correct. We have improved our wording on lines 64-67 as shown in our response to comment 7.

9. ***Comment:*** *Lines 99-100. I think when one is discussing the deployment of simple AOD (or PM2.5) sensors, the common concern was a tradeoff between their cheapness (and possibility to create a very dense sampling network) and their notorious instability especially in high relative humidity conditions or just variable weather conditions. When use of low-cost sensor network is proposed, one should always mention about their accuracy or the tradeoff between accuracy and easiness/cheapness/simplicity.*

**Response:** We have edited text on lines 98-102 to acknowledge the tradeoffs associated with using low-cost sensors as follows:

"Samplers used in these networks must be sufficiently low-cost to deploy in large numbers, have manageable operational and maintenance requirements, and provide useful and reliable PM2.5 and AOD measurements (i.e., measurement data of sufficient accuracy and precision so as to support scientific inference or public decision-making). Thus, consideration should be given to the tradeoffs associated with deploying low-cost sensors such as scalability and simplicity versus accuracy and reliability."

10. ***Comment:*** *Line 109. More information about also systematic noise, instrumental error and sensors stability (all this information determined in work #1 or #2) should be explicitly provided here. This is a problem for the entire article as it supposed to be a strong point, but actually ends up as a weak one.*

**Response:** We have added more validation information as on lines 109-117 as follows:

"The results of a field validation campaign revealed agreement to within 10% (mean relative error) for AOD values relative to co-located AERONET instruments. The mean AOD difference was <0.01 with 95% confidence upper and lower limits of agreement of 0.03 and -0.02, respectively. With respect to PM2.5, the AMODv1 filter measurements agreed within 8% (mean relative error) relative to Federal Equivalent Method (FEM) monitors from the Environmental Protection Agency (EPA), with a mean difference of -0.004 µg m-3 and 95% confidence upper and lower limits of agreement of 1.84 and -1.85 µg m-3, respectively (Wendt et al., 2019). With respect to real-time PMS5003 PM2.5 measurements, the mean relative error between the AMODv1 and an FEM monitor was 1.98 µg m-3 with and mean difference of 0.04 µg m-3 and 95% confidence upper and lower limits of agreement of 5.02 and -4.95 µg m-3, respectively (Wendt et al., 2019)."

11. ***Comment:*** *Line 133. What about accuracy of a sensor? This attribute is implied in "mechanical Robustness"?*

**Response:** Sensor accuracy was indeed a key design consideration for the AMODv2. We have updated lines 141-143 to include accuracy as a design priority as follows:

"Thus, we prioritized a design that is low-cost, accurate, mechanically robust, portable, automated, and user-friendly. We provide images of AMODv2 hardware in Fig. 1, highlighting key internal and external components."

12. **Comment:** *Line 266-267 Once again I have a feeling of advertisement.*

**Response:** We have re-worded this sentence to describe the necessary setup materials in a more neutral tone as follows on lines 281-283:

"Materials needed to initiate a sample include an AMODv2 monitor, a cartridge loaded with a pre-weighed filter, and a smartphone with the AMODv2 control application installed ("CEAMS"; available on the Apple App Store and Google Play)."

13. **Comment:** *Line 279. Is temporal variability-driven uncertainty of average AOD (PM2.5) reported for a user or for a reader of this article?*

**Response:** If we are understanding your question correctly, yes, each raw AOD and PM2.5 measurement is available in addition to the average value. In other words, the AMODv2 outputs (and logs) all individual PM2.5 (or individual AOD) measurements taken over each 3-minute sampling interval. To simplify the website interface, we limit data presented there to the 3-minute average values, but at the end of each sample, the AMODv2 pushes the full data file so the user can analyse the uncertainty of the average measurements. We have added the following text on lines 300-302 to clarify what information is available to the user.

"Individual measurements of AOD and PM2.5, from which averages are derived, are available in the complete file, facilitating post-sample uncertainty analysis of PM2.5 and AOD measurements."

14. **Comment:** *Line 288. It would be nice to have information about the distances of these sensors between each other. They are collocated? It is confusing because the authors say "we collocated our instruments within 50 m" but it is not clear whether AERONET-sensor couples were collocated (I guess they don't have 10 sun photometers) or sensor-to-sensor collocation was made based on 50 m distance between them. This collocation criteria should be more clearly articulated; some table of sensor locations or map would be useful otherwise.*

**Response:** All AMODv2 instruments were co-located with each other, within 50 m of a single reference instrument. We have updated the paragraph in question for clarity on lines 304-311 as follows:

"We assessed precision and bias of AMODv2 AOD sensors relative to an AERONET monitor at the NEON-CVALLA site in Longmont, Colorado (40.160 N, 105.167 W) between June 2020 and December 2020. We co-located our instruments within 50 m of the reference instrument (and within 5 m of each other) on nine separate days with varying atmospheric conditions (e.g. wildfire smoke and clean air) using a total of 14 unique AMODv2 units. Each test consisted of 2 to 4 hours of sampling at a rate of one sample approximately every 3 minutes. The AERONET reference monitor sampled at a frequency of one sample approximately every 15 minutes. AMODv2 and AERONET measurements concurrent within 2 minutes were included in the validation data set. The accuracy of AMODv2 AOD measurements was assessed via Deming regression."

15. **Comment:** *Line 301. Any reference to these wildfires? What is the argument led to conclusion that the instrument was measuring exactly smoke during their analysis?*

**Response:**
We have moved this section from the main text to the supplement. Our measurements were performed during the Cameron Peak fire, the largest wildfire in Colorado's history. The following plot shows the location of the Colorado State University Powerhouse campus where the measurements were taken and the location of the fires and the HMS smoke plumes for October 16-17, 2020.

[Figure]

16. **Comment:** *Line 307. I think the statement about accuracy of AMODv2 is hasty without providing details about their stability.*

**Response:** We have removed section 3.1 from the main text of the manuscript. It is now available in supplement 1. We have restructured the manuscript to include the following sections, which we believe more directly support the research objectives of this manuscript:
- 3.1 Summary and table of design improvements
- 3.2 Co-located reference validation
- 3.4 Reliability analysis
- 3.5 Potential sampler applications

Regarding the statement in question, we have removed reference to accuracy in this supplement. The introduction to this portion of the supplement now reads as follows:

"Here we present results from the sample deployment of 10 units. We configured the units to sample for approximately 60 hours."

17. **Comment:** *Figure 3, are these realistic spikes in PM2.5? 300 μg/m-3. What is the uncertainty of this spike? 30 μg/m-3 or more? How realistic are temporally averaged estimates of PM2.5 during such peaky periods of high aerosol load? Is such strong temporal variability reflected in some parameter for a user of this sensor or (more importantly) for a reader of this scientific article? I cannot distinguish most points; I see mostly magenta points. Why the authors use so visually inconvenient color scale? There are no black or gray points but I see two types of magenta points that are undistinguishable. All points have kind of pastel tones making it even worse, blue looks like red, red looks like magenta, etc.*

**Response:** Although we have moved this section to the supplement, we will address the reviewer's comments here. Plantower values have been shown to correlate well with reference monitors, but the values are biased high at high concentrations, as has been specifically shown for wildfire events (e.g. Holder et al., 2020, Sayahi et al., 2019). The Plantower sensor manual states that the sensor is effective from 0-500 $\mu g/m^3$; although Holder et al. (2020) suggests that the sensor signal may become saturated over 200 $\mu g/m^3$. Studies have shown that comparisons improve with longer time averages and with the application of a correction factor (e.g. Holder et al., 2020, Sayahi et al., 2019; Delp and Singer, 2020; Mehadi et al., 2020; Barkjohn et al., 2020). The Plantower sensor itself has an internal correction factor that reduces concentrations by ⅔ (PM25_ATM), but most studies suggest using the raw PM25_CF1 (e.g., Tryner et al., 2020), which we use here (to note, there has been some confusion with regards to PurpleAir monitors use of CF1 and ATM values from the Plantower as these were previously incorrectly labeled). In Barkjohn et al. (2020), they found that the Plantower sensors in PurpleAir (purpleair.com) monitors could overestimate daily concentrations by 40%, but using a correction factor could

reduce the bias in the 24-hour average to 3 ug/m$^3$. This correction factor is used by the fire.airnow.gov website. Holder et al. (2020) focused specifically on evaluation and development of a correction factor for wildfire smoke. They found a MAE of 66.2 μg/m$^3$ (NRMSE of 143.3%) for hourly Plantowers values compared to reference monitors during wildfire smoke events. By applying a smoke-specific correction factor, they reduced the MAE to 7.61 μg/m$^3$ (NRMSE of 16.9%). While many studies suggest using a correction factor, this is always done in post-processing so that users can choose the appropriate correction factor (4 are currently available for application from a dropdown menu on the PurpleAir website). There are "ambient air" correction factors and source-specific correction factors (i.e., wildfire smoke as mentioned here and woodsmoke correction factors). The evaluation and development of correction factors for Plantowers is a growing research topic. In Part 2, we corrected the Plantower with the filter measurements as per Tryner et al. (2020); however, discussion of the appropriate method or correction factor to use for different air masses seemed outside the scope of this paper, and thus we just displayed uncorrected 5-minute average Plantower data in this plot. The literature does support that the uncertainty on the peak in our plot could be on the order of 30 μg/m$^3$ or more.

While we acknowledge that the Plantower values are biased high; a 300 ug/m$^3$ spike for a 5-minute average is not unrealistic for this period compared to FEM monitors along the Front Range in Colorado as shown in the following plot. The Longmont monitoring station (AQS ID: 080130003) had hourly average concentration of over 300 ug/m$^3$, Boulder (AQS ID: 080131001) had hourly concentrations over 150 ug/m$^3$, Greeley (AQS ID: 081230006) had an hourly concentration of 100 ug/m$^3$, and the Fort Collins station (AQS ID: 080690009) had hourly averages of almost 100 ug/m$^3$.

[Figure]

With respect to the color scale, we have changed the color scale and transparency to improve the readability of the plot in Figure S7.

18. **Comment:** *Line 319. Are there any other arguments confirming the "moderate air pollution event" in the area except AMODv2 that is being tested and evaluated in this study?*

**Response:** We have removed the original section 3.1. In the supplementary section, we have added uncertainty values to the $PM_{2.5}$ concentrations measured by AMODv2 based on our work in AMODv1. Also see comment 17 for more information on the $PM_{2.5}$ levels during this period.

19. **Comment:** *Line 329. Once again, the website is mentioned, what does it give to a reader that data from the sample deployment were accessed from this website? Redundant information.*

**Response:** We have removed mention of the website in the supplement.

20. **Comment:** *Line 332. Statement about improvement of AMODv2 wireless connection. There should be some kind of table whereas technical characteristics (and superiority) of AMODv2 that are not related to atmospheric measurement techniques are directly shown versus inferior AMODv1.*

**Response:** We have removed the original section 3.1. Information regarding technical improvements to the AMODv1 are now summarized in Table 1 as follows:

.**Table 1: Design comparison between AMODv1 and AMODv2**

| Design specification | AMODv1 | AMODv2 |
|---|---|---|
| Sample interval | 48 hours | 120 hours |
| Sample flow rate | 2 L min$^{-1}$ | 1 L min$^{-1}$ |
| Sun alignment procedure | Manual using pinhole aperture target | Automatic dual-axis closed-loop sun tracking system |
| AOD cloud screening | None available | Automatic AOD triplet measurement screening protocol |
| AOD measurement frequency | 1 measurement per day | 1 measurement every 20 minutes during daytime hours |
| Data logging | MicroSD card | MicroSD card, wireless data transfers every 20 minutes, and complete file wireless data transfer at the end of each sample |
| Data visualization | None available | Real-time $PM_{2.5}$ and AOD plots on website |
| Real-time debugging information | None available | Sample flow rate, total sampled volume, battery temperature, battery |

| | | voltage, state of charge, current draw, and wireless signal strength |
|---|---|---|
| Manufacturing Cost | $1,100 | $1,175 |

21. **Comment:** *Line 333. "Weatherproof design" The authors should either provide arguments for extensive evaluation of these sensors in variable weather conditions proving that sensors are actually weatherproof or move this description for the aforementioned table. In the first case, a test for low and high temperatures, variable humidity conditions and mixed aerosol conditions should be shown. Several geographic locations are desirable as we cannot conclude about such stability just based on observations in Colorado. In the case if the authors decide to make weatherproof evaluation, I foremost recommend to check whether their PM2.5 concentrations are subjected to influence of high relative humidity as shown in Crilley et al., (2020) whereas the authors have noticed hygroscopic growth of aerosols under >60% RH Conditions.*

**Response:** We have removed the original section 3.1. Here "weatherproof" refers to the mechanical and electrical components being protected from weather damage as an outdoor sampler. We have removed the term "weatherproof" in this context in the supplementary material to reduce confusion.

22. **Comment:** *Line 335. Once again, some redundant details of technical improvements of sensors in the journal dedicated to atmospheric measurement techniques. "Data accessibility" can be also moved to the table with comparison of AMODv1 and AMODv2. Also, the description of this paragraph is purely qualitative. If some engineers want to check the key indicators of stability of these sensors, they cannot because there are no such quantitative data here. If atmospheric scientists read this, once again, redundant qualitative information for the main text of the journal like AMT.*

**Response:** We have eliminated the redundant information and summarized relevant design changes in Table 1.

23. **Comment:** *Lines 354-356. Only biomass burning event is analyzed here or also clean conditions Included?*

**Response:** The summary statistics are calculated on the full range of validation data, including data under clean conditions and during the biomass burning event. We have added the following text on lines 363-369 as follow:

"Summary statistics calculated on the full set of measurement pairs across all measurement conditions are provided for each wavelength in Table 2.

**Table 2: Summary statistics for AMODv2 vs. AERONET co-located tests**

| Wavelength (nm) | Mean absolute error (AOD) | Deming slope coefficient | $R^2$ | AOD Precision (AOD) |
|---|---|---|---|---|
| 440 | 0.04 | 0.953 | 0.987 | 0.02 |
| 500 | 0.06 | 0.985 | 0.978 | 0.03 |
| 675 | 0.03 | 1.011 | 0.995 | 0.01 |
| 870 | 0.03 | 1.015 | 0.977 | 0.02 |

Summary statistics on the data set partitioned into clear and elevated-AOD samples are presented in Table S1. The definitions of clear and elevated-AOD samples are explained in the description of Table S1."

24. ***Comment:*** *Table 1. AOD precision and mean absolute error are finally provided here. Systematic uncertainties should be reported when AOD estimates are reported throughout the Manuscript.*

**Response:** With the revised manuscript and supplement, AOD estimates are reported with appropriate uncertainty values. Note that the AOD ranges reported for the validation experiments are from AERONET instruments. Therefore the uncertainties are given as ± 0.01, as stated in Giles et al., 2019**. This is stated on lines 331-332 as follows:

"AOD values reported by AERONET during validation experiments ranged from $0.035 \pm 0.01$ to $1.59 \pm 0.01$ at 440 nm, $0.030 \pm 0.01$ to $1.51 \pm 0.01$ at 500 nm, $0.021 \pm 0.01$ to $1.13 \pm 0.01$ at 675 nm, and $0.016 \pm 0.01$ to $0.77 \pm 0.01$ at 870 nm."

25. ***Comment:*** *Line 360. What about mean absolute errors depending on the environmental conditions, they varied from biomass burning event and clean conditions?*

**Response:** These data are provided in supplementary Table S1, which is also replicated below. Briefly, the AOD mean absolute errors of the AMODv2 were found to be largely independent of the AOD magnitude.

**Table S1:** AMODv2 validation summary statistics calculated separately for elevated-AOD days and clear days. Elevated-AOD days were defined as days in which the average AERONET AOD at 500 nm was greater than or equal to 0.15. Clear days were defined as days in which the average AOD was less than 0.15. In total, five days were identified as clear and four days were identified as elevated-AOD.

| Wavelength (nm) | AERONET 500 nm AOD magnitude | Number of samples | Mean absolute error (AOD) | AOD Precision (AOD) |
|---|---|---|---|---|
| 440 | All included | 426 | 0.04 | 0.02 |
| 500 | All included | 426 | 0.06 | 0.02 |
| 675 | All included | 426 | 0.03 | 0.01 |
| 870 | All included | 426 | 0.03 | 0.02 |
| 440 | > 0.15 (elevated) | 115 | 0.05 | 0.02 |
| 500 | > 0.15 (elevated) | 115 | 0.05 | 0.02 |
| 675 | > 0.15 (elevated) | 115 | 0.03 | 0.01 |
| 870 | > 0.15 (elevated) | 115 | 0.03 | 0.01 |
| 440 | < 0.15 (clear) | 311 | 0.04 | 0.02 |
| 500 | < 0.15 (clear) | 311 | 0.06 | 0.03 |
| 675 | < 0.15 (clear) | 311 | 0.02 | 0.01 |
| 870 | < 0.15 (clear) | 311 | 0.03 | 0.02 |

26. *Comment: Figure 4. "Fort Lupton, CO" I think it's better to explain CO as Colorado if mentioned for non-American readers.*

**Response:** We have corrected the figure caption to read as follows:

"Figure 3: AERONET (MAXAR-FUTON site in Fort Lupton, Colorado, USA) vs. AMODv2 AOD co-located comparison (n=426) results with panels separated by wavelength. Lines of best fit were calculated via Deming regression analysis."

27. *Comment: Line 389 "was explained primarily by the constant term" that is one of the reasons why uncertainties should be explicitly shown.*

**Response:** With the revised manuscript and supplement, AOD estimates are reported with appropriate uncertainty estimates.

28. ***Comment:*** *Line 391 Once again it's unclear whether only biomass burning event or mixed events are Implied?*

**Response:** The mean-difference plot is for the full, mixed data set. We have clarified this in the caption for what is now Fig. 5 as follows:

"Mean-difference plot for measurements taken by AERONET and AMODv2 monitors, with panels separated by wavelength. Paired AERONET and AMODv2 under both clear and biomass burning conditions (as defined in Table S1) are included."

29. ***Comment:*** *Line 404. and again, instrumental errors due to optical sensor drift over time should be timely quantified in the manuscript to be used for AOD estimates the authors report for analyzing biomass burning event.*

**Response:** We have replaced this statement with an analysis of calibration stability in the section at the end of Section 3.2 as follows:

"Previous work has noted the tendency for optical interference filters to degrade over time, changing the accuracy of the most recent calibration (Brooks and Mims, 2001; Giles et al., 2019). We quantified the long-term stability of the AMODv2 AOD sensors by re-calibrating 16 AMODv2 units 15 months after their initial calibration. Summary statistics quantifying the change calibration constant (V0) changes are provided in Table 3."

**Table 3: Summary statistics for AMODv2 calibration stability test. All summary statistics refer to the change in $V_0$ (Eq. 2). Note that the absolute value of the maximum change refers to the single unit with the highest percent change for each wavelength.**

| Wavelength (nm) | Average absolute value of change (%) | Median change (%) | Absolute value of maximum change (%) |
|---|---|---|---|
| 440 | 13.84 | -7.14 | 62.72 |
| 500 | 11.80 | -9.64 | 37.08 |
| 675 | 6.66 | -0.75 | 29.40 |
| 870 | 14.63 | -2.80 | 50.72 |

A plot illustrating the voltage change undergone by each of the 16 AMODv2 units is provided in Fig. 6.

[Figure]

**Figure 6: Linear change plots illustrating the change to $V_0$ (Eq. 2) from the initial calibration to a follow up test calibration of 16 AMODv2 units. Starting and ending calibration voltage values are provided on the vertical axis. Panels are separated by wavelength. Each line represents the change after 15 months of a single wavelength channel of an AMODv2 unit.**

"The results presented in Fig. 6 illustrate that the calibration constants (V0 in Eq. 2) remained relatively stable (changes of 5% or less) for most AMODv2 units over the course of 15 months. However, several units exhibited relatively large changes (in excess of 30%) in their calibration constants, indicating calibration changes may vary considerably by unit. Boersma and de Vroom (2006) present theoretical analyses and conclude that the calculation of AOD is most sensitive to errors in the calibration constant, V0. Their theoretical analyses combined with the results in Fig.

6, point to drift in V0 as a likely source for large, unit specific errors in AOD AMODv2 measurements. To limit errors due to calibration drift, we recommend that AMODv2 V0 values be re-calibrated on an annual basis. Determining the source of changes to the calibration constants of some AMODv2 units is the subject of ongoing investigation. Potential sources include changes to the photodiode sensor element, the optical interference filters, and the protective glass window element in the light path of the sensors."

30. ***Comment:*** *Line 407. How large are discrepancies in the production dates between these sensors actually to result to such differences?*

**Response:** We have replaced this statement with an analysis of calibration stability in section 3.2. See our response to comment 29.

31. ***Comment:*** *Line 411. Denoting discussion as a subsection is rather uncommon.*

**Response:** We have removed this section.

32. ***Comment:*** *Lines 420-433. Technical improvements such as data accessibility, protocol, time resolution of observations without indications how it can improve the agreement with the referenced AOD/PM2.5 observations or fill the existing gaps in this field are really minor contribution for atmospheric sciences given the existence of works #1 and #2 on this topic.*

**Response:** We have removed this paragraph as much of its information is now present uniquely in section 3.1.

33. ***Comment:*** *Line 427. Price of a sensor is an important information but given the lack of really useful conclusions for atmospheric science, the price reference just looks odd in discussion.*

**Response:** See our response to comment 32, we have removed this section. Price is now listed in Table 1.

34. ***Comment:*** *Line 434-439 Other types of cheap AOD sensors can be also (and already being) used for this purpose. The authors should nail down the implications for citizen science provided particularly by improvement version of AMOD (v2) since the articles about AMODv1 were already present.*

**Response:** The unsupervised nature of AMODv2 samples, which allows it to take multiple AOD measurements throughout the day, is the primary advantage of AMODv2 over AMODv1 with respect to citizen science. We have highlighted this point on lines 496-499 as follows:

"The new design of the AMODv2 allows for unsupervised measurement and quality control protocols that reduce the operational demands on a study volunteer, particularly compared with AMODv1 and other low-cost AOD sensors. Deployments with citizen scientists are ongoing and data from those campaigns will be the subject of future studies."

35. **Comment:** *Line 445 "Data collection that would normally require multiple instruments is possible with a single AMODv2 unit" this sentence requires explanation.*

**Response:** This sentence has been removed.

36. **Comment:** *Line 446. Once again, there are other instruments and also cheap sensors that have been already exploited for this purpose (with less than a minute temporal sampling of PM2.5), so what's new here?*

**Response:** We have removed the sentence in question.

37. **Comment:** *Line 479. Proposal to use AMODv2 in citizen science is repetitive.*

**Response:** We have removed former lines 470 through 483.

38. **Comment:** *Line 484. The conclusions look like discussion about the scientific findings of this article that should be extended. Meanwhile, the two latest paragraphs of the discussion look like conclusions of this article, which are however, not based on the findings about evident advantages of AMODv2 for atmospheric science, but on minor technical improvements of this type of sensor.*

**Response:** We have extended the conclusions section to include current and future applications of the AMODv2.

39. **Comment:** *Line 490. The proposal about establishing network for AOD monitoring is repetitive given the information provided in the discussion. The authors are advised to thoroughly check which information they will use as discussion and which as conclusion. This should be done for avoiding logical loops and repetitions.*

**Response:** We have removed this statement in the updated conclusions section.

**Response Bibliography**

Ford, B., Pierce, J. R., Wendt, E., Long, M., Jathar, S., Mehaffy, J., Tryner, J., Quinn, C., van Zyl, L., L'Orange, C., Miller-Lionberg, D., and Volckens, J.: A low-cost monitor for measurement of fine particulate matter and aerosol optical depth – Part 2: Citizen-science pilot campaign in northern Colorado, Atmospheric Meas. Tech., 12, 6385–6399, https://doi.org/10.5194/amt-12-6385-2019, 2019.

Giles, D. M., Sinyuk, A., Sorokin, M. G., Schafer, J. S., Smirnov, A., Slutsker, I., Eck, T. F., Holben, B. N., Lewis, J. R., Campbell, J. R., Welton, E. J., Korkin, S. V., and Lyapustin, A. I.: Advancements in the Aerosol Robotic Network (AERONET) Version 3 database – automated near-real-time quality control algorithm with improved cloud screening for Sun photometer aerosol optical depth (AOD) measurements, Atmospheric Meas. Tech., 12, 169–209, https://doi.org/10.5194/amt-12-169-2019, 2019.

Sayahi, T., Butterfield, A., and Kelly, K. E.: Long-term field evaluation of the Plantower PMS low-cost particulate matter sensors, Environ. Pollut., 245, 932–940, https://doi.org/10.1016/j.envpol.2018.11.065, 2019.

Tryner, J., L'Orange, C., Mehaffy, J., Miller-Lionberg, D., Hofstetter, J. C., Wilson, A., and Volckens, J.: Laboratory evaluation of low-cost PurpleAir PM monitors and in-field correction using co-located portable filter samplers, Atmos. Environ., 220, 117067, https://doi.org/10.1016/j.atmosenv.2019.117067, 2020.

Wendt, E. A., Quinn, C. W., Miller-Lionberg, D. D., Tryner, J., L'Orange, C., Ford, B., Yalin, A. P., Pierce, J. R., Jathar, S., and Volckens, J.: A low-cost monitor for simultaneous measurement of fine particulate matter and aerosol optical depth – Part 1: Specifications and testing, Atmospheric Meas. Tech., 12, 5431–5441, https://doi.org/10.5194/amt-12-5431-2019, 2019.

---

## Author Comment (AC2)

We thank the reviewers for their insightful comments, which have allowed us to produce a stronger manuscript. Our responses to the general and specific comments are given below. Reviewer comments are provided in italics and our responses are given in plain text. Line number references pertain to the revised manuscript with no tracked changes. Quoted text from the revised manuscript is given in blue in this response document.

**Reviewer 2:**

Reviewer 2 general comments

**Comment:** *The authors present improvements in AMOD sensing device for concurrent PM2.5 and AOD measurements. They describe a sample deployment of 10 AMODv2s for 2.5 days during a wildfire smoke event, and they present a 7-day validation campaign of the AMODv2 that was co-located with a AOD reference device. The results indicate close agreement between AMODv2s and the reference instrument although they identified some intra-device variability. They conclude that the AMODv2 is well suited for citizen science and other high-spatial-density deployments due to its low cost, compact form, user-friendly interface, and high measurement frequency of AOD and PM2.5. The paper addresses an important topic, a cost-effective strategy for improving satellite estimates of PM2.5 concentration. They present results to support their statements about measurement accuracy. However, it presents rather limited data to support many of the authors' other conclusions. The paper would be strengthened with additional data. It also requires some clarification regarding real-time presentation of PM2.5 concentrations as well as some date inconsistencies. If additional results are not possible, the manuscript may be more suitable as a technical note.*

**Response:** Thank you for your suggestions. Since our initial submission, we have added several new results and analyses that address concerns outlined in this review. The specific details are provided in our responses to the relevant reviewer comments.  We feel that this work is sufficient to qualify for a standalone research paper, as per the aims and scope of *Atmospheric Measurement Techniques*.  However, if the managing editor for AMT feels that this work is better suited as a technical note, we will be pleased to have it considered as such.

Reviewer 2 general comments

Significant weaknesses

1. **Comment:** *Limited scope of data. The website presented in the paper illustrates that the AMODv2s are collecting and presenting measurements to the public, which is great. It is puzzling why the authors present only 2.5 days of measurements (2 days of AOD measurements)*

*from 14 devices. This limited data set also makes it difficult to support the authors statements that the system is autonomous.*

**Response:** Field campaigns are currently ongoing and results from those studies will be presented in separate papers. The original section 3.1 was included solely as a demonstration of the AMODv2 capabilities (apart from lab testing). That section has now been moved to the supplement in favor of new results and analyses (field deployments were put on hold in 2020 due to the COVID-19 pandemic). The measurements taken by participants require more context and analysis that we felt was outside the goals of this paper. However, we have added additional data to the manuscript as detailed in our subsequent responses.

2. ***Comment:*** *Presented longer-term measurements, which is one of the key motivations for this developing autonomous measurements. The paper would be greatly strengthened if they:*

**Response:** In Section 3.3, we present a summary of samples collected between January and March of 2021, for the purpose of analyzing the ability of the AMODv2 to complete its intended sample consistently. Specifically, this section includes failure rate analysis on 76 test runs conducted on the roof of our laboratory building. Measurements were taken on three separate weeks. We describe this experiment in the manuscript on lines 317-325 as follows:

"We tested the reliability of AMODv2 instruments in a series of 5-day, outdoor samples on the roof of a Colorado State University laboratory facility (430 N College Avenue, Fort Collins, Colorado, USA). All units were co-located within a 10 m radius. We started tests on January 16, 2021, January 30, 2021, and March 31, 2021, which included 34, 27, and 15 unique AMODv2 units respectively, for a total of 76 samples. We assessed the reliability of the AMOD according to the rate at which samples terminated prematurely. Samples that failed to reach at least 115 hours of the intended 120 hour sample duration were designated as premature terminations. We specifically assessed the mechanical robustness of AMODv2 units by visually inspecting failed units for evidence of water ingress and electrical component damage. We also analyzed the AOD data from these samples to evaluate the automatic solar alignment procedure and quality control algorithm."

To clarify, automation in this article refers specifically to AOD measurement and data transfer. The AMODv2 is not autonomous in the sense that it can be left unattended for longer than than the programmed five-day sample. After each sample, an operator needs to replace the gravimetric filter cartridge and charge the unit for ~8 hours. In other words, each discrete five-day sample is completed unsupervised, as described in Figure 2. However, after the device is fully charged, it can be re-deployed, enabling sampling in consecutive weeks. At the time of writing, 31 AMODv2 units are currently deployed across the USA in a citizen-science network. Our participants are starting samples every Tuesday for a span of 8 weeks starting in mid-June

2021. The results of this study are outside of the scope of this article and will be detailed in a future work.

3. **Comment:** *Co-located a few AMODv2s to understand intra-device variability, and have long-term validation measurements by co-locating the AMODv2s with AOD reference devices for a long time period (seasons to a year). The manuscript provides some evidence that device to device variability could be important.*

**Response:** The final portion of what is now Section 3.2 includes calibration stability analysis of 16 AMODv2 units. The description of this experiment is given on lines 312-316 as follows:

"We evaluated the long-term stability of the AOD sensors by re-calibrating a set of 16 AMODv2 units 15 months after their initial calibration. Original calibrations for the units tested were conducted at the MAXAR-FUTON site in Fort Lupton, Colorado, USA (40.036 N, 104.885 W) on February 21, 2020.  Re-calibrations were conducted at the NEON-CVALLA site on May 27, 2021 (The MAXAR-FUTON site was indefinitely unoperational at the time of the second calibration)."

The results of this experiment are provided on lines 418-444 as follows:

Previous work has noted the tendency for optical interference filters to degrade over time, changing the accuracy of the most recent calibration (Brooks and Mims, 2001; Giles et al., 2019). We quantified the long-term stability of the AMODv2 AOD sensors by re-calibrating 16 AMODv2 units 15 months after their initial calibration.  Summary statistics quantifying the change calibration constant (V0) changes are provided in Table 3.

**Table 3: Summary statistics for AMODv2 calibration stability test. All summary statistics refer to the change in $V_0$ (Eq. 2). Note that the absolute value of the maximum change refers to the single unit with the highest percent change for each wavelength.**

| Wavelength (nm) | Average absolute value of change (%) | Median change (%) | Absolute value of maximum change (%) |
|---|---|---|---|
| 440 | 13.84 | -7.14 | 62.72 |
| 500 | 11.80 | -9.64 | 37.08 |
| 675 | 6.66 | -0.75 | 29.40 |
| 870 | 14.63 | -2.80 | 50.72 |

 A plot illustrating the voltage change undergone by each of the 16 AMODv2 units is provided in Fig. 6.

[Figure]

**Figure 6: Linear change plots illustrating the change in calibration voltage, $V_0$ (Eq. 2), from the initial calibration to a follow up test calibration of 16 AMODv2 units. Each instrument is represented by a separate line with starting and ending calibration voltage values delineated on the vertical axis. Panels are separated by wavelength. Each line represents the change after 15 months of a single wavelength channel of an AMODv2 unit.**

The results presented in Fig. 6 illustrate that the calibration constants (V0 in Eq. 2) remained relatively stable (changes of 5% or less) for most AMODv2 units over the course of 15 months. However, several units exhibited relatively large changes (in excess of 30%) in their calibration constants, indicating calibration changes may vary considerably by unit. Boersma and de Vroom (2005) present theoretical analyses and conclude that the calculation of AOD is most sensitive to errors in the calibration constant, V0. (Boersma and de Vroom, 2006). Their theoretical analyses

combined with the results in Fig. 6, point to drift in V0 as a likely source for large, unit specific errors in AOD AMODv2 measurements. To limit errors due to calibration drift, we recommend that AMODv2 V0 values be re-calibrated on an annual basis. Determining the source of changes to the calibration constants of some AMODv2 units is the subject of ongoing investigation. Potential sources include changes in sensitivity or drift of the photodiode sensor element, degrading of the optical interference filters, and/or clouding of the protective glass window element in the light path of the sensors.

4. **_Comment:_** _Presented data that indicates the autonomous nature of the device, % of device up time, and number of valid AOD measurements (quality control described with triplicate measurements), number of AOD measurements excluded for cloud cover, etc. This needs to be done for weeks or months of measurements to begin to support the autonomous claims in the manuscript._

**Response:** In what is now Section 3.3 we have added an analysis that includes 76 samples (each 5 days in duration) completed at our laboratory rooftop. There we provide failure, and AOD quality control results as follows:

"AMODv2 sensor validation results from this work and prior work indicate that the instrument can accurately measure AOD and PM2.5 when operating properly. However, for effective large-scale deployments, AMODv2 units must reliably complete their intended sampling protocol when deployed outdoors for 120 hours. Potential causes of premature sample failure included, premature battery drainage, damage to mechanical or electrical components (e.g. water ingress into motors or sensors), and firmware related crashes (e.g. memory overflow errors). In a series of reliability tests on the rooftop of our laboratory facility, we found that of 76 attempted samples, 75% were successfully completed, 16% failed due to premature battery drainage, 8% failed due to water damage, and 1% (one unit) failed due to a firmware crash. To address failures due to premature battery drainage, we replaced batteries that would not fully charge and replaced motors that were drawing excess current. To address failures due to water damage, we replaced damaged boards and applied additional sealant to key mechanical interfaces. We addressed the firmware crash issue by reconfiguring the memory allocation to grant more memory to the wireless data push functionality, which proved to be the most memory intensive sub-system. Overheating was not an issue in the testing discussed here, as the testing was conducted in winter months. We will test the AMODv2 under warmer conditions to evaluate heating effects on the performance of the instrument.

We also verified that AMODv2 units were attempting AOD measurements and applying the prescribed data screening protocols. In the 76 test samples, AMODv2 units attempted 22,419 AOD measurements per wavelength. Units detected the sun and took at least one measurement toward forming a triplet 4,763 times per wavelength. The results partitioned by quality control designation are provided in Table 3. Instances where an AMODv2 reported a numerical AOD

value were considered valid AOD measurements. Instances where an AMODv2 failed to acquire three AOD measurements for a single measurement sequence (Fig. S6) were designated as incomplete with a unique error code. Cloud-screened measurements were those where the solar alignment is achieved for 3 measurements but the triplet failed to meet the acceptance criteria (Fig. S6).

**Table 3: Results from the AMODv2 quality control algorithm from 4.763 AOD measurements taken in laboratory rooftop testing. Attempts where zero measurements were logged for a triplet attempt are omitted from the table.**

| Wavelength (nm) | Proportion of valid AOD measurements | Proportion of invalid AOD measurements | |
|---|---|---|---|
| | | Incomplete AOD triplets | Cloud-screened measurements |
| 440 | 33% | 20% | 46% |
| 500 | 34% | 20% | 45% |
| 675 | 35% | 20% | 44% |
| 870 | 33% | 20% | 46% |

The results of this study indicate the AMODv2 automatically acquired solar alignment for a complete measurement triplet on 80% of attempted measurements. However, among the completed triplets, approximately 45% of measurements were identified as cloud-contaminated and subsequently screened. The screening algorithm did not reach consistent results across all wavelengths, as evident by slight deviations in the proportion of screened data across wavelengths. In this work, we applied the same exclusion criteria to each wavelength (Fig. S6). These results indicate unique exclusion criteria may be necessary for each wavelength to achieve consistent results, particularly when there is substantial deviation in magnitude between two measurement wavelengths (e.g. 440 nm AOD much higher than 870 nm AOD for a single measurement)."

**5. *Comment:*** *The validation measurements of AMODv2 are also limited. The authors state that their validation measurements cover a range of conditions, which is important. If the AMODv2 is capable of autonomous measurement, it is unclear why the validation results are so limited. It is also unclear how the authors came to 426 data paired points. I would encourage them to include a table in the supplementary materials that presents the device ID, the number of valid AOD measurements each day, the number and % of excluded measurements (if any), % of missing data and the cloud and PM2.5 conditions of the day. I struggled with understanding how 426 paired data points were achieved. If the sampling rate was 2.5 to 3 minutes, are they presenting*

*the results of each test (one every two to three hours, this would get closer to 426 data points were the samples averaged? ) or samples selectively included.*

**Response:** We respectfully disagree that the AOD validation data set is limited. Compared with our AMODv1 work, our AMODv2 work includes 296 more measurement pairs per measurement wavelength (Wendt et al., 2019). We also had limited opportunities to measure at high AOD due to the transient nature of wildfire smoke and the generally clean air in Colorado. Therefore, most additional data we could have collected would have been at low AOD. We had already shown in the AMODv1 paper that our AOD system performs well at low AOD (Wendt et al., 2019). Additional data at low AOD would have left the two ends of the AOD spectrum unbalanced, in terms of the number of measurements, affecting our regression analysis in particular.

There were also logistical challenges that limited our access to AERONET monitors. Owing to the high cost (>$55,000) of AERONET Cimel sun photometers we cannot purchase one of our own to perform AOD validation. We traveled one hour by car to reach an AERONET site at Vance Brand Airport in Longmont, Colorado. We could not access the private area where the AERONET photometer is kept due to COVID-19 restrictions. Instead, we set up our units at a public table outside the fence but still relatively near the monitor, where it was not safe to leave the units unattended for extended periods of time.

The device-specific AOD data requested by the reviewer are available in the data set associated with the article (amod_v2_validation.csv) available at the following link: https://hdl.handle.net/10217/225291. There were no measurements excluded due to clouds. We selected days with no cloud cover to perform our validation experiments. Also, a team member was always there to observe that none of the measurements were cloud-contaminated.

As for the 426 number, the AERONET instrument measured at a lower frequency (1 measurement every 15 minutes) than AMODv2 units, which were custom-configured to sample at the higher rate of 2.5 to 3 minutes specifically for validation. This means that many AMODv2 measurements were omitted from the data set because they lacked an accompanying coincident AERONET measurement. We have added lines 308-309 to the manuscript to clarify this point.

“The AERONET reference monitor sampled at a frequency of one sample approximately every 15 minutes.”

6. **Comment:** *With less than 10 days of data total (2.5 of sample deployment and 7 days of validation), it is difficult to see how this annual calibration recommendation is reached. Line 408: "To mitigate unit specific errors, we recommend re-calibrating instruments at least one time per year." PM2.5 concentrations. The paper presents PM2.5 measurements from the Plantower PMS5003. The authors need to clarify if they are presenting measurements that were*

*corrected with their on-board filter measurements (or if these are raw measurements or corrected using some other strategy). They should provide these filter-based correction factors in the supplementary material. If they are using some other method, they should describe this. They should also describe if they are applying corrections to Plantower's raw ATM or CF=1 measurements. Also, in several places in the manuscript, the authors discuss real-time visualization and data access. However, as the authors and others have published previously the Plantower PMS 5003 raw measurements are biased. They require a correction factor for the aerosol conditions of interest. Reasonably accurate real-time PM2.5 concentration estimates would not be possible if the filter cartridges require one week of sampling, must be mailed in, and weighed. These corrections would only be possible weeks after the PM2.5 measurements are posted. The authors need to clarify what they are presenting and if the measurements are corrected how they are corrected.*

**Response:** As a minor correction, we conducted nine, co-located AOD evaluation experiments, not seven. We have provided this detail in lines 305-307 as follows:

"We co-located our instruments within 50 m of the reference instrument (and within 5 m of each other) on nine separate days with varying atmospheric conditions (e.g. wildfire smoke and clean air) using a total of 14 unique AMODv2 units."

The statement regarding re-calibration is now supported by the analysis presented at the end of Section 3.2 (see response to comment 3).

We have moved this discussion of the wildfire event and the plots to the supplement. In Figure S8, we show the Plantower PMS5003 PM25_CF1 rather than the PM25_ATM, following Tryner et al. (2020) and our Ford et al. (2019, Part 2 of this series). However, both the CF1 and ATM values are output and stored on the internal log file. The Plantower has a widely known bias, and many groups have developed correction factors for the Plantower. These correction factors are applied in post-processing and have generally been developed for longer averaging periods (i.e., the Barkjohn et al. 2020 was developed for 24-hour average concentrations) rather than 5-minute averages. The application of the correction factor is an open area of research (by our group and others); thus, we have decided to display the raw values in real-time on our website as does the PurpleAir website (purpleair.com), whose sensors include the same Plantower model. All participants are informed that the data displayed on the website is preliminary (AOD values have a simple cloud screening applied for the website, but, as in Part 2, we perform more QA/QC in post-processing). From our multiple deployments this year, we plan to further investigate developing correction factors from our concurrent filter and Plantower measurements that could be applied in real-time for future AMOD deployments (our datasets now cover a wider range of locations and atmospheric conditions). We have edited the text to state that these values are the raw CF1 values in Figure S8. "PM$_{2.5}$ measurements are from the Plantower PMS5003 and are the

CF = 1 values. These values have not been corrected using the PMS5003 atmospheric correction factor, but were not corrected relative to the filter mass concentrations."

We have moved Section 3.1 to the supplement and clarified that $PM_{2.5}$ measurements were not corrected using the PMS5003 atmospheric correction factor, and are not corrected relative to the filter.

Moderate weaknesses

7. **Comment:** *Table 1. What types of conditions are being presented here. All 7 days or a subset? How do the statistics vary by condition?*

**Response:** Table 1 summary statistics include the full data set with no separation by condition. Partitioned data are provided in supplementary Table S1, which is also replicated below. Briefly, the AOD mean absolute errors of the AMODv2 were found to be largely independent of the AOD magnitude.

**Table S1:** AMODv2 validation summary statistics calculated separately for elevated-AOD days and clear days. Elevated-AOD days were defined as days in which the average AERONET AOD was greater than or equal to 0.15. Clear days were defined as days in which the average AOD at 500 nm was less than 0.15. In total, five days were identified as clear and four days were identified as elevated-AOD.

| Wavelength (nm) | AERONET 500 nm AOD magnitude | Number of samples | Mean absolute error (AOD) | AOD Precision (AOD) |
|---|---|---|---|---|
| 440 | All included | 426 | 0.046 | 0.023 |
| 500 | All included | 426 | 0.056 | 0.027 |
| 675 | All included | 426 | 0.026 | 0.0089 |
| 870 | All included | 426 | 0.033 | 0.017 |
| 440 | > 0.15 (elevated) | 115 | 0.054 | 0.019 |
| 500 | > 0.15 (elevated) | 115 | 0.051 | 0.019 |
| 675 | > 0.15 (elevated) | 115 | 0.027 | 0.006 |
| 870 | > 0.15 (elevated) | 115 | 0.031 | 0.007 |
| 440 | < 0.15 (clear) | 311 | 0.043 | 0.024 |
| 500 | < 0.15 (clear) | 311 | 0.059 | 0.030 |
| 675 | < 0.15 (clear) | 311 | 0.025 | 0.010 |

| 870 | < 0.15 (clear) | 311 | 0.034 | 0.021 |

8. ***Comment:*** *The manuscript in places reads a little like an advertisement for the AMODv2. The authors should focus on statements that are supported by their results (i.e., the terms autonomous, realtime, user friendly, use for citizen science). See comments about autonomous and real-time in the previous section.*

**Response:** We have identified and altered or removed portions of the manuscript that read like an advertisement. Specifically, we have removed the discussion section and moved statements of current and future applications into the conclusions.

9. ***Comment:*** *Figure S7 does not occur during either the sample deployment (October 2021 or maybe 2020, see comment on date inconsistencies below) or during the co-location validation testing. It is unclear why the authors would select a time other than the sample deployment described in the paper. If there is a good reason, they should provide it. They should also discuss the causes of the PM2.5 concentration peaks. Also, this figure illustrates that more than one half of the AMODv2s are offline (see comment about autonomous).*

**Response:** The snapshot in Figure S7 was taken at a time when AMODv2 units were located at different locations in Colorado for test deployments, for purposes of illustrating the web interface. The validation data were taken with the samplers located together. With co-located samplers, points on the map would overlap and obscure what information is available on the web interface. We have clarified the purpose of the figure and the presence of inactive circles in the caption as follows:

[Figure]

"Sample live map from sampler website csu-ceams.com overlaid with time series of $PM_{2.5}$. This snapshot was taken at a time when AMODv2 units were located at different locations in Colorado for test deployments, for purposes of illustrating the web interface. Colored circles represent active AMODv2s. Grey circles represent inactive AMODv2 units. Inactive circles are typically units that have been assigned a location on the map, but are at a location with poor Wi-Fi connectivity, in between samples, or recently sent back by the operator. The color scale is determined by the current Air Quality Index (AQI) calculated based on the $PM_{2.5}$ measurement. The four sample $PM_{2.5}$ time series plots are linked to specific participant locations with arrows. Time series plots can be accessed by clicking on an active circle. Users may select the option to view AOD from a drop-down menu for both the map and the time series plot. Note: that this figure has been edited to show map and time series plots on the same page. On the actual website selecting a point displays only one simplified time series on the map itself. Detailed time series shown here are available on a separate page which can be accessed through selecting a unit on the map."

10. ***Comment:*** *Given that the AMODv2 requires WiFi, wouldn't it be reasonable to have a plug in option, which would likely increase up-time, see comment regarding figure S7.*

**Response:** We are currently testing a plug in option as part of a collaboration with NASA's Jet Propulsion Laboratory, as they are interested in long-term deployment of the AMOD to support future satellite remote sensing missions. The intention is to achieve longer runtimes with less

operator interaction. Depending on the success of these tests, we may transition the standard operation protocol to support continuous line power. However, for now, five day samples on battery power remain the default mode of operation.

11. **Comment:** *It is unclear why the authors present only 2.5 days of results when the devices are capable of 5 days of measurements.*

**Response:** Note that this section has been moved to the supplement. The units deployed were not fully charged before the sample. We did not realize this until after the samples terminated. We chose to include this test due to the broad range of air pollution levels covered in a short period of time. The original intention of this section was to highlight the advantages in automatic AOD data collection compared with the one-sample-per-day paradigm of the AMODv1. We analyze the ability of AMODv2 units to complete five-day samples in Section 3.3.

12. **Comment:** *This may be just my eyes, but Figure 4 doesn't look like it contains 426 data points.*

**Response:** We have verified that each panel in what was formerly Figure 4 contains 426 data points. Due to close agreement of AMODv2 units with other AMODv2 units, there is a high level of overlap of the data points, particularly at low AOD. We direct the reviewer to our open-access dataset that will be archived with the manuscript (https://hdl.handle.net/10217/225291).

13. **Comment:** *The Figure 2 shows one sampling interval rather than continuous sampling. The filter cartridge needs to be replaced and the instrument restarted. This should be included in the figure.*

**Response:** We have updated the caption to clarify the flowchart applies to a single sample as follows:

"Figure 2: Overall device operation flow diagram for a single sample. After each sample, the AMODv2 must be recharged for at least eight hours before a new sample can be started. Manual inputs require operator intervention. Automatic processes are executed with no operator intervention. Predefined processes are detailed in Supplemental Figs. S1-S6. Parallel processes are executed pseudo-simultaneously using a real-time operating system."

14. **Comment:** *Line 288 The authors discuss their sample deployment of co-located 10 AMOD units during a wildfire smoke event in Fort Collins, Colorado in October of 2021. I believe that they mean October 2020 since October 2021 has not yet occurred.*

**Response:** We have corrected the dates in the relevant supplementary material text.

15. ***Comment:*** *Line 374 The authors should clarify how they derived their expected error equations.*

**Response:** We have added lines 385-389 to further describe the equations:

"We derived expected error (EE) equations to constrain the error of AMODv2 measurements relative to AERONET as a function of AOD (following the form used in the validation of satellite AOD products compared to AERONET AOD). We derived the equations iteratively by adjusting the constant and linear terms until the bounds defined by Eqs. (4) through (7) each contained 85% of the co-located measurement pairs for each wavelength."

16. ***Comment:*** *Line 427. The authors are not comparing apples to apples here. First, they are comparing their component costs to commercial instruments, which are likely 5 to 10X the component costs.*

**Response:** We have removed this cost comparison.

17. ***Comment:*** *Further the GRIMM EDM is a Federal Reference Method. A more reasonable comparison would be a low-cost particulate matter sensor, i.e., Purple Air at $250 or something like the TSI DustTrak $5000 with a cyclone cut point and filters that can be weighed.*

**Response:** We have removed this cost comparison.

Minor
***Comment:*** *Line 395 "by wavelength" is repeated*

**Response:** We have removed the repeated text.

**Response Bibliography**

Ford, B., Pierce, J. R., Wendt, E., Long, M., Jathar, S., Mehaffy, J., Tryner, J., Quinn, C., van Zyl, L., L'Orange, C., Miller-Lionberg, D., and Volckens, J.: A low-cost monitor for measurement of fine particulate matter and aerosol optical depth – Part 2: Citizen-science pilot campaign in northern Colorado, Atmospheric Meas. Tech., 12, 6385–6399, https://doi.org/10.5194/amt-12-6385-2019, 2019.

Giles, D. M., Sinyuk, A., Sorokin, M. G., Schafer, J. S., Smirnov, A., Slutsker, I., Eck, T. F., Holben, B. N., Lewis, J. R., Campbell, J. R., Welton, E. J., Korkin, S. V., and Lyapustin, A. I.: Advancements in the Aerosol Robotic Network (AERONET) Version 3 database – automated near-real-time quality control algorithm with improved cloud screening for Sun photometer aerosol optical depth (AOD) measurements, Atmospheric Meas. Tech., 12, 169–209, https://doi.org/10.5194/amt-12-169-2019, 2019.

Sayahi, T., Butterfield, A., and Kelly, K. E.: Long-term field evaluation of the Plantower PMS low-cost particulate matter sensors, Environ. Pollut., 245, 932–940, https://doi.org/10.1016/j.envpol.2018.11.065, 2019.

Tryner, J., L'Orange, C., Mehaffy, J., Miller-Lionberg, D., Hofstetter, J. C., Wilson, A., and Volckens, J.: Laboratory evaluation of low-cost PurpleAir PM monitors and in-field correction using co-located portable filter samplers, Atmos. Environ., 220, 117067, https://doi.org/10.1016/j.atmosenv.2019.117067, 2020.

Wendt, E. A., Quinn, C. W., Miller-Lionberg, D. D., Tryner, J., L'Orange, C., Ford, B., Yalin, A. P., Pierce, J. R., Jathar, S., and Volckens, J.: A low-cost monitor for simultaneous measurement of fine particulate matter and aerosol optical depth – Part 1: Specifications and testing, Atmospheric Meas. Tech., 12, 5431–5441, https://doi.org/10.5194/amt-12-5431-2019, 2019.

---

## Referee Report (RR1)

**1. Original Submission**

**1.1. Recommendation**

Accept after minor revision

**2. General comments:**

amt-2021-73

*"A low-cost monitor for simultaneous measurement of fine particulate matter and aerosol optical depth – Part 3: Automation and design improvements"*

**Overall opinion:** This version has been markedly improved. I appreciate the tailored amendments and modifications that were applied based on my comments to this article. I think the research design, the results and the supplementary information are satisfactory for publication. However, abstract and conclusions may need further minor polishing. I am pretty sure that after this, the article can be published. Good luck!

**2.1. Specific comments:**

1. **Abstract.** You can mention that PM2.5 not just impact public health, but **NEGATIVELY** impact public health. You have better description of this aspect in the first sentences of the introduction. Check, please. Also, you need to find a way to include three ideas in the 2nd sentence of the abstract: measuring aerosols is important, there are many networks and instruments to do it, but there are still gaps and that 's why cheap sensor technology is required. The concluding sentence (or 2 sentences) about "why this research is important for science/industry/society" is missing. Please try formulating these sentences. For instance
   * the improved sensors can be deployed for citizen science efforts in the cities where aerosol observations are scarce (no reference), but weather conditions are variable (inferior cheap sensors would suffer from excessive instability).
   * another option, you can check WMO (World Meteorological Organization) requirements for AOD measurements' accuracy. As your sensor is very precise and stable, you can state that you introduce the method+sensor that meet WMO requirements for AOD measurements and therefore can objectively qualify as a nominee as the core for new global-scale network for AOD measurements in future.
2. **Conclusions:** My previous comment (1st stage of revision) about the discussion was aimed to show that it is uncommon to denote a discussion as a subsection, but you do not necessarily need to delete it completely. You can call your conclusion section "Discussion and conclusions" in the present form also. From my point of view, it would be logical to make just 2 paragraphs of the "Discussion and conclusions". First paragraph may consist of what is described between lines 500 and 515. The second paragraph may consist of what is the first paragraph of the conclusions now (Lines 488-499). As mentioned above, I suggest adding as tailored as possible implication that logically stems from your work. Likewise in the intro, you can add the concluding sentence (or 2 sentences) about "why this research is important for science/industry/society". These sentences can be identical for intro and conclusions. This recommendation is important because smartly tailored implications help scientists (a) to

conclude whether your work results can lay the basis for a next series of similar research, and also may increase (b)"citability" of your article later.

**2.2. Minor comments:**

3. Line 42. "We present results from a trial development aimed at assessing…" Too complex sentence, simplify to "we conducted trial development and assessed…". Something like this.
4. Lines 513… "Such networks could provide…" This sentence is undesirable because, how this information can advance satellite remote sensing technologies? It is better to state that this technology will close existing gaps of the global aerosol measurement infrastructure (that is currently based on combination of ground-based and satellite observations).
5. Line 253. 'Unique error code'. In aerosol measurements, this kind of procedure is usually denoted as 'flagging', i.e. 'unique error code' = 'flag'.
6. Check if Levy Zamora reference should be cited as "Levy-Zamora et al., 2019" (with '-') or not, please.
7. Double-check if you included all required AERONET acknowledgments please
8. Table 1. You mention "manufacturing cost". Please mention the date at which this estimate was applicable because the cost may change in future.
9. Line 377. "AOD units" I am not sure if it is correct to refer to the unitless quantity like this.
10. Line 439. Check if you need subscript index in $V_0$ or not. Here and elsewhere, please
11. Supplementary material, to quote the sentence: "with all units reporting consistent values for AOD (>0.3 +/- 0.06)". Please be consistent with precision reporting, check elsewhere.

---

## Author Response (AR2)

We are pleased to learn that our revisions were well received. We appreciate the additional suggestions to strengthen the manuscript. Here we describe the minor revisions we have made to the manuscript, as requested by the reviewer, prior to publication. Reviewer comments are provided in italics and our responses are given in plain text. Line number references pertain to the revised manuscript with no tracked changes. Quoted text from the revised manuscript is given in blue in this response document. We have provided revised versions of both the manuscript and the SI along with accompanying documents.

Specific comments

1. **Comment:** *Abstract. You can mention that PM2.5 not just impact public health, but NEGATIVELY impact public health. You have better description of this aspect in the first sentences of the introduction. Check, please. Also, you need to find a way to include three ideas in the 2nd sentence of the abstract: measuring aerosols is important, there are many networks and instruments to do it, but there are still gaps and that 's why cheap sensor technology is required. The concluding sentence (or 2 sentences) about "why this research is important for science/industry/society" is missing. Please try formulating these sentences. For instance*
   - *the improved sensors can be deployed for citizen science efforts in the cities where aerosol observations are scarce (no reference), but weather conditions are variable (inferior cheap sensors would suffer from excessive instability).*
   - *another option, you can check WMO (World Meteorological Organization) requirements for AOD measurements' accuracy. As your sensor is very precise and stable, you can state that you introduce the method+sensor that meet WMO requirements for AOD measurements and therefore can objectively qualify as a nominee as the core for new global-scale network for AOD measurements in future.*

**Response:** We edited the first sentence of the abstract on lines 21-22 as follows:

"Atmospheric particulate matter smaller than 2.5 micrometers in diameter (PM2.5) has a negative impact on public health, the environment, and Earth's climate."

We have edited the second sentence of the abstract on lines 22-25 as follows:

"Consequently, a need exists for accurate, distributed measurements of surface-level PM2.5 concentrations at a global scale. Existing PM2.5 measurement infrastructure provides broad PM2.5 sampling coverage, but does not adequately characterise community-level air pollution at high temporal resolution. This motivates the development of low-cost sensors which can be more practically deployed in spatial and temporal configurations currently lacking proper characterization."

We have added the following conclusive sentences on lines 48-51 as follows:

"We demonstrate that the AMODv2 is an accurate, stable and low-cost platform for air pollution measurement. We describe how the AMODv2 can be implemented in spatial citizen-science networks where reference-grade sensors are economically impractical and low-cost sensors lack accuracy and stability."

2. *Comment: Conclusions. My previous comment (1ˢᵗ stage of revision) about the discussion was aimed to show that it is uncommon to denote a discussion as a subsection, but you do not necessarily need to delete it completely. You can call your conclusion section "Discussion and conclusions" in the present form also. From my point of view, it would be logical to make just 2 paragraphs of the "Discussion and conclusions". First paragraph may consist of what is described between lines 500 and 515. The second paragraph may consist of what is the first paragraph of the conclusions now (Lines 488-499). As mentioned above, I suggest adding as tailored as possible implication that logically stems from your work. Likewise in the intro, you can add the concluding sentence (or 2 sentences) about "why this research is important for science/industry/society". These sentences can be identical for intro and conclusions. This recommendation is important because smartly tailored implications help scientists (a) to conclude whether your work results can lay the basis for a next series of similar research, and also may increase (b)"citability" of your article later.*

**Response:** We have titled the final section "Discussion and conclusions" and re-ordered the paragraphs according to the reviewers suggestions.

With respect to the ending sentences of the introduction and conclusions sections, we have added the following text at the end of of the introduction on lines 141-144 as follows:

"The results presented here demonstrate that AMODv2 is a practical option to establish spatially-dense PM2.5 and AOD measurement networks. Applied in these networks, the AMODv2 will close gaps in the existing global aerosol measurement infrastructure of ground-based and satellite-based observations."

And the conclusions on lines 513-516 as follows:

"The portability, performance, and low cost of the AMODv2 make it a practical option to establish spatially-dense PM2.5 and AOD measurement networks. Applied in these networks, the AMODv2 will close gaps in the existing global aerosol measurement infrastructure of ground-based and satellite-based observations."

Minor comments

3. **Comment:** *Line 42. "We present results from a trial development aimed at assessing…" Too complex sentence, simplify to "we conducted trial development and assessed…". Something like this.*

**Response:** The sentence on lines 45-46 now reads as follows:

"We conducted a trial deployment to assess the reliability and mechanical robustness of AMODv2 units."

4. **Comment:** *Lines 513… "Such networks could provide…" This sentence is undesirable because, how this information can advance satellite remote sensing technologies? It is better to state that this technology will close existing gaps of the global aerosol measurement infrastructure (that is currently based on combination of ground-based and satellite observations).*

**Response:** We have edited the sentence on lines 142-144 as follows:

"Applied in these networks, the AMODv2 will close gaps in the existing global aerosol measurement infrastructure of ground-based and satellite-based observations."

5. **Comment:** *Line 253. 'Unique error code'. In aerosol measurements, this kind of procedure is usually denoted as 'flagging', i.e. 'unique error code' = 'flag'.*

**Response:** We have altered the wording to clarify the flags are numerical to add more information with error code and flag being synonymous. The edited text is as follows:

"Measurements identified as cloud-contaminated are flagged with a unique numerical code. Measurements with incomplete triplets are also flagged with a unique numerical code."

6. **Comment:** *Check if Levy Zamora reference should be cited as "Levy-Zamora et al., 2019" (with '-') or not, please.*

**Response:** We have verified that the author's name is not hyphenated.

7. **Comment:** *Double-check if you included all required AERONET acknowledgments please.*

**Response:** We have reviewed the document and verified necessary AERONET acknowledgements are present

8. ***Comment:*** *Table 1. You mention "manufacturing cost". Please mention the date at which this estimate was applicable because the cost may change in future.*

**Response:** We have added the date of original manufacturing in the table as follows:

"Manufacturing Cost (As of July 2019)."

9. ***Comment:*** *Line 377. "AOD units" I am not sure if it is correct to refer to the unitless quantity like this.*

**Response:** We have removed the phrase "AOD units" on lines 385-386 as follows:

"With respect to stability across AOD magnitude, the mean absolute error deviated by less than 0.011 between clear days and elevated-AOD days across all wavelengths (Table S1)."

10. ***Comment:*** *Line 439. Check if you need subscript index in V0 or not. Here and elsewhere, please.*

**Response:** We have reviewed the document and ensured all mentions of the variable are italicized with the zero subscripted (i.e. $V_0$).

11. ***Comment:*** *Supplementary material, to quote the sentence: "with all units reporting consistent values for AOD (>0.3 +/- 0.06)". Please be consistent with precision reporting, check elsewhere.*

**Response:** We have reviewed the supplementary material to ensure AOD values are reported with the proper number of digits after the decimal point.